# Language Models Can Learn from Verbal Feedback Without Scalar Rewards

## Abstract

LLMs are often trained with RL from human or AI feedback, yet such methods typically *compress nuanced feedback into scalar rewards*, discarding much of their richness and inducing scale imbalance. We propose treating verbal feedback as a conditioning signal. Inspired by language priors in text-to-image generation, which enable novel outputs from unseen prompts, we introduce the **feedback-conditional policy (FCP)**. FCP learns directly from response-feedback pairs, approximating the feedback-conditional posterior through maximum likelihood training on *offline* data. We further develop an *online bootstrapping* stage where the policy generates under positive conditions and receives fresh feedback to refine itself. This reframes feedback-driven learning as conditional generation rather than reward optimization, offering a more expressive way for LLMs to directly learn from verbal feedback.

## 1 Introduction

> *"That all of what we mean by goals and purposes can be well thought of as maximization of the expected value of the cumulative sum of a received scalar signal (reward)."* — Reward Hypothesis by Richard Sutton

The *reward hypothesis* in reinforcement learning (RL) was proposed over two decades ago (Sutton, 2004), when feedback from the environment had to be reduced to **scalar rewards** for RL algorithms to operate. This view shaped much of the field's progress and remains the prevailing standard in applying RL to *alignment* and *reasoning* for large language models (LLMs) (Ziegler et al., 2019; Bai et al., 2022; Rafailov et al., 2023; Guo et al., 2025).

Yet in practice, the feedback encountered in RL for LLMs, especially in *non-verifiable* settings, is most often **verbalized**, such as "`Good start, but the code can be more efficient`". Such feedback may come from human users (Stephan et al., 2024), generative reward models (Zhang et al., 2024; Mahan et al., 2024), or tool outputs in agentic scenarios (Wang et al., 2025b; Jin et al., 2025). Reducing the verbal feedback into scalar rewards introduces several limitations:

**I. Information loss.** Scalar rewards capture far less information than verbal feedback/critiques and are often uninterpretable. For example, the critiques "`The response is redundant but correct`" and "`The response is compact but has many typos`" may both collapse to a reward of $0.8$, despite describing very different response patterns. Furthermore, the verbalized thoughts produced by (generative) reward models are typically discarded as intermediate outputs, with only the final scalar retained for RL training.

**II. Ambiguity.** Verbal feedback, especially from human users, is often *mixed* (containing both pros and cons), *emotional*, or *uncertain*, such as "`I'm so happy`" or "`I'm not sure, maybe try again?`". Such feedback is far more common than purely positive or negative signals and carries diverse cues for learning and for understanding user interaction styles. Mapping these forms of feedback to scalars could be unclear or arbitrary.

**III. Imbalanced reward scales across tasks.** In multi-task training (e.g., math, code, science, games), it is difficult to maintain a consistent reward scale. Positive feedback on a simple math problem is far easier to obtain than on a challenging coding or game-playing task, which induces imbalanced rewards across domains and biases the learning process.

**Training phase**: Collecting (offline) responses from $\pi_{\text{ref}}$ and feedback from $p_{\text{env}}$ | **Test phase**: Users generate responses from $\pi_\theta$

```python
def flatten(lst):
    return sum(lst, [])
```

$$o_1 \sim \pi_{\text{ref}}(\cdot|x)$$

The code is **compact and simple**, but **only handles one-level nesting**

$$c_1 = (c_1^+, c_1^-) \sim p_{\text{env}}(\cdot|x, o_1)$$

```python
def flatten(lst):
    result = []
    for item in lst:
        if isinstance(item, list):
            result.extend(flatten(item))
        else:
            result.append(item)
    return result
```

$$o_2 \sim \pi_{\text{ref}}(\cdot|x)$$

The code **works for any nesting**, but **is somewhat verbose**

$$c_2 = (c_2^+, c_2^-) \sim p_{\text{env}}(\cdot|x, o_2)$$

The code is **compact and simple**, and **works for any nesting**

$$c^+ = (c_1^+, c_2^+) \text{ is expected by users}$$

```python
def flatten(lst):
    return [
        x
        for e in lst
        for x in (
            flatten(e) if isinstance(e, list) else [e]
        )
    ]
```

$$o^+ \sim \pi_\theta(\cdot|x, c^+)$$

Figure 1: **Learning from mixed verbal feedback.** The instruction $x$ is "*Write a Python function flatten(lst) that returns a flat list of integers*". The reference policy $\pi_{\text{ref}}$ may assign low probability to the ideal response $o^+$, making purely positive response-feedback pairs $(o^+, c^+)$ rare in the training data collected from $\pi_{\text{ref}}$ and $p_{\text{env}}$. This resembles the setting of text-to-image generation, where the *language prior* enables models to combine seen captions (analogous to mixed feedback $c_1$ and $c_2$) and generate rare images (analogous to purely positive response $o^+$) such as "*a banana surfing on the ocean*" (Figure 4). Motivated by this, our model $\pi_\theta$ is trained as a feedback-conditional policy (FCP), and when conditioning on user-defined positive $c^+$, there is $\pi_\theta(o|x, c^+) \propto \pi_{\text{ref}}(o|x) \cdot p_{\text{env}}(c^+|x, o)$.

Scalarization has long been seen as unavoidable, bridging verbal feedback and the numerical signals required by RL. With the rise of large-scale language pretraining, however, this view is being re-examined (Yao, 2025). LLMs embody strong commonsense and linguistic **priors**, suggesting a new paradigm: *treat verbal feedback as a first-class training signal*, rather than forcing it into a scalar form.

After all, LLMs already show the ability to **implicitly understand verbal feedback**. In agentic tasks, they iteratively adapt by integrating feedback prompts from human users, external critiques, or tool calls into their context and refining their responses accordingly (Wang et al., 2025b; Novikov et al., 2025). This indicates that LLMs can process verbal feedback, but only implicitly, *through a latent "mental model" that does not convert understanding into explicit scalar rewards*. The key question, then, is how to distill such feedback into training so that it directly improves model performance, rather than relying on inefficient multi-turn trial and error at test time.

To this end, we propose to learn a **feedback-conditional policy (FCP)** $\pi_\theta(o|x, c) \propto \pi_{\text{ref}}(o|x) \cdot p_{\text{env}}(c|x, o)$, where $\pi_{\text{ref}}(o|x)$ is a reference policy that generates a response $o$ given an instruction $x$, and $p_{\text{env}}(c|x, o)$ is the distribution of environment feedback $c$. Intuitively, the FCP reweighs the reference policy by how likely each response $o$ would elicit the observed feedback $c$. Conditioning on positive feedback $c^+$ gives $\pi_\theta(o|x, c^+) \propto \pi_{\text{ref}}(o|x) \cdot p_{\text{env}}(c^+|x, o)$, which increases the probability of generating responses that are more likely to receive favorable feedback. In this way, the FCP learns a *posterior* distribution that integrates prior knowledge from $\pi_{\text{ref}}$ with verbal feedback, allowing it to handle diverse forms of feedback, including mixed ones, as illustrated in Figure 1.

After training an *offline* FCP $\pi_\theta(o|x, c) \propto \pi_{\text{ref}}(o|x) \cdot p_{\text{env}}(c|x, o)$ that conditions on arbitrary feedback $c$, we further improve it through *online bootstrapping*. Concretely, we conduct online training by sampling rollouts from the behavior policy $\pi_\theta(o|x, c^+)$ (goal-conditioned on positive feedback), and re-annotating them with fresh feedback from $p_{\text{env}}$, thereby iteratively strengthening the policy.

Our pilot experiments show that FCP matches or surpasses strong scalar-based baselines such as offline RFT (Dong et al., 2023) and online GRPO (Shao et al., 2024), **without relying on verifiers, scalar conversion, or data filtering**. This demonstrates a simple and scalable framework that preserves the richness of verbal feedback while avoiding the scarcity of rule-based verifiers and the risk of reward hacking. While our current implementation is naive, advanced training techniques could further improve FCP's performance.

## 2 LEARNING DIRECTLY FROM VERBAL FEEDBACK

Traditional RL methods train a policy by up-weighting responses that receive "good" feedback and down-weighting those that receive "bad" feedback. From a probabilistic view, RL can be seen as learning a *posterior* over responses that are expected to receive good feedback (i.e., high rewards) (Peters & Schaal, 2007; Peng et al., 2019; Rafailov et al., 2023). Distinguishing what counts as good or bad typically requires carefully designed reward functions or detailed rubrics to produce scalar signals, leading to the limitations discussed in Section 1.

Our approach is inspired by language priors in text-to-image generation, where models compose *unseen* prompts from mixed captions (Figure 4). Similarly, language priors could enable LLMs to absorb diverse verbal feedback and yield high-quality responses beyond scalar reinforcement (Figure 1). Since LLMs already show implicit feedback understanding, we train directly on it: *offline* to initialize a **feedback-conditional policy (FCP)** (Section 2.1), then *online* to bootstrap performance (Section 2.2).

## 2.1 OFFLINE TRAINING: INITIALIZING FEEDBACK-CONDITIONAL POLICY

We begin with a reference policy model $\pi_{\text{ref}}$ that takes an input instruction $\boldsymbol{x}$ and generates a response $\boldsymbol{o} \sim \pi_{\text{ref}}(\cdot|\boldsymbol{x})$. The response $\boldsymbol{o}$ then undergoes a *single-turn* interaction with the environment, which provides verbal feedback $\boldsymbol{c} \sim p_{\text{env}}(\cdot|\boldsymbol{x}, \boldsymbol{o})$. The reference policy $\pi_{\text{ref}}$ may represent a base model, an instruction-tuned model, or a reasoning model, and the response $\boldsymbol{o}$ can include both thinking processes and the final answer. The environment $p_{\text{env}}$ may consist of human users or generative reward models. In the **offline** setting, where responses are collected from $\pi_{\text{ref}}$, we define the joint distribution of response-feedback pairs as $P_{\text{off}}(\boldsymbol{o}, \boldsymbol{c}|\boldsymbol{x}) \triangleq \pi_{\text{ref}}(\boldsymbol{o}|\boldsymbol{x}) \cdot p_{\text{env}}(\boldsymbol{c}|\boldsymbol{x}, \boldsymbol{o})$, from which we derive the *feedback-conditional posterior* distribution:

$$P_{\text{off}}(\boldsymbol{o}|\boldsymbol{x}, \boldsymbol{c}) = \frac{P_{\text{off}}(\boldsymbol{o}, \boldsymbol{c}|\boldsymbol{x})}{P_{\text{off}}(\boldsymbol{c}|\boldsymbol{x})} = \frac{\pi_{\text{ref}}(\boldsymbol{o}|\boldsymbol{x}) \cdot p_{\text{env}}(\boldsymbol{c}|\boldsymbol{x}, \boldsymbol{o})}{\sum_{\boldsymbol{o}} \pi_{\text{ref}}(\boldsymbol{o}|\boldsymbol{x}) \cdot p_{\text{env}}(\boldsymbol{c}|\boldsymbol{x}, \boldsymbol{o})}. \tag{1}$$

Informally, let $\boldsymbol{c}^+$ denote purely positive feedback and $\boldsymbol{c}^-$ purely negative one. Mixed feedback can be approximated as $\boldsymbol{c} = (\boldsymbol{c}^+, \boldsymbol{c}^-)$, while neutral or uncertain feedback may be neither. If we condition on positive feedback $\boldsymbol{c}^+$, for instance, "The generated code is functionally correct, efficient, and compact" for a coding instruction $\boldsymbol{x}$, then $P_{\text{off}}(\boldsymbol{o}|\boldsymbol{x}, \boldsymbol{c}^+) \propto \pi_{\text{ref}}(\boldsymbol{o}|\boldsymbol{x}) \cdot p_{\text{env}}(\boldsymbol{c}^+|\boldsymbol{x}, \boldsymbol{o})$, which favors responses $\boldsymbol{o}$ that are more likely to elicit positive feedback.[1]

While $P_{\text{off}}(\boldsymbol{o}|\boldsymbol{x}, \boldsymbol{c}^+) \propto \pi_{\text{ref}}(\boldsymbol{o}|\boldsymbol{x}) \cdot p_{\text{env}}(\boldsymbol{c}^+|\boldsymbol{x}, \boldsymbol{o})$ appears to be the oracle policy we are seeking, it cannot be directly sampled from, because $p_{\text{env}}(\boldsymbol{c}^+|\boldsymbol{x}, \boldsymbol{o})$ is defined only after the full response $\boldsymbol{o}$ is generated, and thus cannot guide generation step by step. We therefore aim to learn a policy that approximates $P_{\text{off}}(\boldsymbol{o}|\boldsymbol{x}, \boldsymbol{c}^+)$. Following Rafailov et al. (2023), we show that $P_{\text{off}}(\boldsymbol{o}|\boldsymbol{x}, \boldsymbol{c}^+)$ is the optimal solution to a KL-constrained reward maximization problem with reward function $\log p_{\text{env}}(\boldsymbol{c}^+|\boldsymbol{x}, \boldsymbol{o})$:

$$P_{\text{off}}(\boldsymbol{o}|\boldsymbol{x}, \boldsymbol{c}^+) \in \arg\max_{\pi} \mathbb{E}_{\pi(\boldsymbol{o}|\boldsymbol{x}, \boldsymbol{c}^+)} \left[\log p_{\text{env}}(\boldsymbol{c}^+|\boldsymbol{x}, \boldsymbol{o})\right] - \mathbb{D}_{\text{KL}}\left(\pi(\boldsymbol{o}|\boldsymbol{x}, \boldsymbol{c}^+)||\pi_{\text{ref}}(\boldsymbol{o}|\boldsymbol{x})\right). \tag{2}$$

In the special case where the environment provides *verifiable rewards*, that is, $p_{\text{env}}(\boldsymbol{c}^+|\boldsymbol{x}, \boldsymbol{o}^+) = 1$ for correct responses $\boldsymbol{o}^+$ and $p_{\text{env}}(\boldsymbol{c}^+|\boldsymbol{x}, \boldsymbol{o}^-) = 0$ for incorrect responses $\boldsymbol{o}^-$, we can show that $P_{\text{off}}(\boldsymbol{o}|\boldsymbol{x}, \boldsymbol{c}^+)$ reduces to the optimal solution of a 0-1 binary reward maximization problem without KL regularization: $P_{\text{off}}(\boldsymbol{o}|\boldsymbol{x}, \boldsymbol{c}^+) \in \arg\max_{\pi} \mathbb{E}_{\pi(\boldsymbol{o}|\boldsymbol{x}, \boldsymbol{c}^+)} \left[\mathbb{1}(\boldsymbol{o} \text{ is } \boldsymbol{o}^+)\right]$ (proof is in Appendix A.1).

**Alternative learning objective.** In more general scenarios, particularly when feedback comes from human users, *solving Eq. (2) is typically intractable*. This is because we can only sample from $p_{\text{env}}$ but cannot compute the exact log-likelihood $\log p_{\text{env}}(\boldsymbol{c}^+|\boldsymbol{x}, \boldsymbol{o})$. Note that the objective in Eq. (2) is equivalent to minimizing the *reverse* KL divergence between $\pi(\boldsymbol{o}|\boldsymbol{x}, \boldsymbol{c}^+)$ and $P_{\text{off}}(\boldsymbol{o}|\boldsymbol{x}, \boldsymbol{c}^+)$:

$$\max_{\pi} \mathbb{E}_{\pi}\left[\log p_{\text{env}}(\boldsymbol{c}^+|\boldsymbol{x}, \boldsymbol{o})\right] - \mathbb{D}_{\text{KL}}\left(\pi(\boldsymbol{o}|\boldsymbol{x}, \boldsymbol{c}^+)||\pi_{\text{ref}}(\boldsymbol{o}|\boldsymbol{x})\right) \Leftrightarrow \min_{\pi} \mathbb{D}_{\text{KL}}\left(\pi(\boldsymbol{o}|\boldsymbol{x}, \boldsymbol{c}^+)||P_{\text{off}}(\boldsymbol{o}|\boldsymbol{x}, \boldsymbol{c}^+)\right),$$

which is derived in Eq. (7). To avoid intractability of computing $\log p_{\text{env}}(\boldsymbol{c}^+|\boldsymbol{x}, \boldsymbol{o})$ in the reverse KL divergence, we instead turn to minimize the *forward* KL divergence between $\pi(\boldsymbol{o}|\boldsymbol{x}, \boldsymbol{c}^+)$ and $P_{\text{off}}(\boldsymbol{o}|\boldsymbol{x}, \boldsymbol{c}^+)$. This relaxation is standard in KL-constrained LM training, where forward-KL objectives have been used for distributional or reward-conditioned generation (Khalifa et al., 2020; Korbak et al., 2022; Pandey et al., 2024), though our setting differs in conditioning on rich verbal feedback. In practice, however, we can only obtain feedback from $p_{\text{env}}(\boldsymbol{c}|\boldsymbol{x}, \boldsymbol{o})$, and it is infeasible to sample exclusively from the constrained subset of positive feedback $p_{\text{env}}(\boldsymbol{c}^+|\boldsymbol{x}, \boldsymbol{o})$ without carefully designed rubrics or filtering. To address this, we generalize the objective: rather than approximating only $P_{\text{off}}(\boldsymbol{o}|\boldsymbol{x}, \boldsymbol{c}^+)$, we learn to approximate $P_{\text{off}}(\boldsymbol{o}|\boldsymbol{x}, \boldsymbol{c})$, conditioning directly on *any* feedback $\boldsymbol{c}$.

Specifically, we propose to learn a **feedback-conditional policy (FCP)** $\pi_\theta(\boldsymbol{o}|\boldsymbol{x}, \boldsymbol{c})$ by minimizing the expected forward KL divergence between $\pi_\theta(\boldsymbol{o}|\boldsymbol{x}, \boldsymbol{c})$ and $P_{\text{off}}(\boldsymbol{o}|\boldsymbol{x}, \boldsymbol{c})$:

$$\min_{\pi_\theta} \mathbb{E}_{P_{\text{off}}(\boldsymbol{c}|\boldsymbol{x})} \left[\mathbb{D}_{\text{KL}}(P_{\text{off}}(\boldsymbol{o}|\boldsymbol{x}, \boldsymbol{c})||\pi_\theta(\boldsymbol{o}|\boldsymbol{x}, \boldsymbol{c}))\right] \Leftrightarrow \max_{\pi_\theta} \mathbb{E}_{P_{\text{off}}(\boldsymbol{c}|\boldsymbol{x})} \left[\mathbb{E}_{P_{\text{off}}(\boldsymbol{o}|\boldsymbol{x}, \boldsymbol{c})} \left[\log \pi_\theta(\boldsymbol{o}|\boldsymbol{x}, \boldsymbol{c})\right]\right]$$
$$\Leftrightarrow \max_{\pi_\theta} \mathbb{E}_{\pi_{\text{ref}}(\boldsymbol{o}|\boldsymbol{x})} \left[\mathbb{E}_{p_{\text{env}}(\boldsymbol{c}|\boldsymbol{x}, \boldsymbol{o})} \left[\log \pi_\theta(\boldsymbol{o}|\boldsymbol{x}, \boldsymbol{c})\right]\right], \tag{3}$$

---

[1] Conditioning on negative feedback $\boldsymbol{c}^-$ would similarly favor poor responses, though this is rarely useful.

---

**Algorithm 1** Offline training: Initializing feedback-conditional policy (Section 2.1)

---

**Inputs:** Reference policy $\pi_{\text{ref}}(\boldsymbol{o}|\boldsymbol{x})$, feedback environment $p_{\text{env}}(\boldsymbol{c}|\boldsymbol{x}, \boldsymbol{o})$, feedback-conditional policy $\pi_\theta(\boldsymbol{o}|\boldsymbol{x}, \boldsymbol{c})$, instruction corpus $\mathcal{X}$, batch size $B$, optimizer $\mathcal{O}$

**Outputs:** The offline-trained parameters $\theta_{\text{off}}$

1: **Initialize** $\boxed{\pi_\theta(\boldsymbol{o}|\boldsymbol{x}, \boldsymbol{c}) = \pi_{\text{ref}}\left(\boldsymbol{o}\,\big|\,\big[\texttt{<EF>}\boldsymbol{c}\texttt{</EF>}, \boldsymbol{x}\big]\right)}$, where $\texttt{<EF>}$ and $\texttt{</EF>}$ are special tokens used to wrap the expected feedback $\boldsymbol{c}$, which is concatenated before the instruction $\boldsymbol{x}$

2: **Collect** offline dataset $\mathcal{D}_{\text{off}} = \{(\boldsymbol{x}, \boldsymbol{o}, \boldsymbol{c})\}$ with $\boldsymbol{x} \sim \mathcal{X}, \boldsymbol{o} \sim \pi_{\text{ref}}(\cdot|\boldsymbol{x})$ then $\boldsymbol{c} \sim p_{\text{env}}(\cdot|\boldsymbol{x}, \boldsymbol{o})$

3: **Objective:** $\boxed{\max_\theta \mathbb{E}_{(\boldsymbol{x}, \boldsymbol{o}, \boldsymbol{c}) \sim \mathcal{D}_{\text{off}}}[\log \pi_\theta(\boldsymbol{o}|\boldsymbol{x}, \boldsymbol{c})]}$      // Taking expectation over $\boldsymbol{x} \sim \mathcal{X}$ in Eq. (3)

4: **while** not converged **do**

5:      Sample $\{(\boldsymbol{x}_i, \boldsymbol{o}_i, \boldsymbol{c}_i)\}_{i=1}^B \sim \mathcal{D}_{\text{off}}$;    $\theta \leftarrow \mathcal{O}.\text{step}\big(\theta, \nabla_\theta \frac{1}{B} \sum_{i=1}^B \log \pi_\theta(\boldsymbol{o}_i|\boldsymbol{x}_i, \boldsymbol{c}_i)\big)$

6: **return** $\theta_{\text{off}} \leftarrow \theta$

---

where the second equivalence follows from the identities $P_{\text{off}}(\boldsymbol{c}|\boldsymbol{x}) \cdot P_{\text{off}}(\boldsymbol{o}|\boldsymbol{x}, \boldsymbol{c}) = P_{\text{off}}(\boldsymbol{o}, \boldsymbol{c}|\boldsymbol{x}) = \pi_{\text{ref}}(\boldsymbol{o}|\boldsymbol{x}) \cdot p_{\text{env}}(\boldsymbol{c}|\boldsymbol{x}, \boldsymbol{o})$. This objective in Eq. (3) reduces to maximum likelihood training, which is straightforward to implement and optimize with data collected from $\pi_{\text{ref}}(\boldsymbol{o}|\boldsymbol{x})$ and $p_{\text{env}}(\boldsymbol{c}|\boldsymbol{x}, \boldsymbol{o})$, as described in Algorithm 1. Its optimal solution is $\boxed{\pi_\theta^*(\boldsymbol{o}|\boldsymbol{x}, \boldsymbol{c}) = P_{\text{off}}(\boldsymbol{o}|\boldsymbol{x}, \boldsymbol{c})}$ on the support set of $P_{\text{off}}(\boldsymbol{c}|\boldsymbol{x})$. Notably, our approach does not require explicitly distinguishing positive $\boldsymbol{c}^+$ from negative $\boldsymbol{c}^-$; the language prior embedded in LLMs can implicitly interpret and combine information from diverse forms of feedback, including mixed ones as seen in Figure 1. At test time, users may specify desired positive feedback $\boldsymbol{c}^+$, and responses can be generated from $\pi_\theta(\boldsymbol{o}|\boldsymbol{x}, \boldsymbol{c}^+)$.

**Remark I: why using $P_{\text{off}}(\boldsymbol{c}|\boldsymbol{x})$?** In Eq. (3), the expectation on $\boldsymbol{c}$ is taken w.r.t. $P_{\text{off}}(\boldsymbol{c}|\boldsymbol{x})$. In principle, any other distribution $p(\boldsymbol{c}|\boldsymbol{x})$ could be used, and the optimal solution $\pi_\theta^*(\boldsymbol{o}|\boldsymbol{x}, \boldsymbol{c}) = P_{\text{off}}(\boldsymbol{o}|\boldsymbol{x}, \boldsymbol{c})$ would remain unchanged on the support $\text{supp}(p(\cdot|\boldsymbol{x}))$. We adopt $P_{\text{off}}(\boldsymbol{c}|\boldsymbol{x})$ mainly for two reasons: **(i)** its support set $\text{supp}(P_{\text{off}}(\cdot|\boldsymbol{x})) = \bigcup_{\boldsymbol{o} \in \text{supp}(\pi_{\text{ref}}(\cdot|\boldsymbol{x}))} \text{supp}(p_{\text{env}}(\cdot|\boldsymbol{x}, \boldsymbol{o}))$ covers all feedback that may be encountered when collecting offline data; **(ii)** it serves as a compensating distribution that converts the intractable posterior expectation $P_{\text{off}}(\boldsymbol{o}|\boldsymbol{x}, \boldsymbol{c})$ into the tractable joint expectation $P_{\text{off}}(\boldsymbol{o}, \boldsymbol{c}|\boldsymbol{x}) = \pi_{\text{ref}}(\boldsymbol{o}|\boldsymbol{x}) \cdot p_{\text{env}}(\boldsymbol{c}|\boldsymbol{x}, \boldsymbol{o})$, which is convenient to sample from.

**Remark II: FCP as inverse dynamics.** We observe that our FCP learning in Eq. (3) aligns with modeling *inverse dynamics* (Brandfonbrener et al., 2023), complementing supervised finetuning (SFT) as *behavior cloning*, and critique finetuning (CFT) (Wang et al., 2025a) as *forward dynamics*. A detailed discussion of this analogy is provided in Appendix A.2.

## 2.2 ONLINE TRAINING: BOOTSTRAPPING BY CONDITIONING ON POSITIVE FEEDBACK

We denote the model obtained by solving the offline problem in Eq. (3) as $\pi_{\theta_{\text{off}}}(\boldsymbol{o}|\boldsymbol{x}, \boldsymbol{c})$, which is capable of generating responses conditioned on any user-defined feedback $\boldsymbol{c}$. Building on this model, we further perform **online training** to bootstrap performance by *conditioning explicitly on positive feedback $\boldsymbol{c}^+$*. Concretely, we iteratively update parameters $\theta_{t+1}$ using rollouts from $\pi_{\theta_t}(\boldsymbol{o}|\boldsymbol{x}, \boldsymbol{c}^+)$ for $t \in \mathbb{N}$, with $\theta_0 = \theta_{\text{off}}$ initialized from the offline solution, as described in Algorithm 2.

Formally, we define the joint distribution $P_{\theta_t}(\boldsymbol{o}, \boldsymbol{c}, \boldsymbol{c}^+|\boldsymbol{x}) \triangleq p_{\text{user}}(\boldsymbol{c}^+|\boldsymbol{x}) \cdot \pi_{\theta_t}(\boldsymbol{o}|\boldsymbol{x}, \boldsymbol{c}^+) \cdot p_{\text{env}}(\boldsymbol{c}|\boldsymbol{x}, \boldsymbol{o})$, where $p_{\text{user}}(\boldsymbol{c}^+|\boldsymbol{x})$ denotes the distribution (fixed or trainable) of user-specified *expected* positive feedback. The corresponding feedback-conditional posterior is

$$P_{\theta_t}(\boldsymbol{o}|\boldsymbol{x}, \boldsymbol{c}) = \frac{P_{\theta_t}(\boldsymbol{o}, \boldsymbol{c}|\boldsymbol{x})}{P_{\theta_t}(\boldsymbol{c}|\boldsymbol{x})} = \frac{\sum_{\boldsymbol{c}^+} p_{\text{user}}(\boldsymbol{c}^+|\boldsymbol{x}) \cdot \pi_{\theta_t}(\boldsymbol{o}|\boldsymbol{x}, \boldsymbol{c}^+) \cdot p_{\text{env}}(\boldsymbol{c}|\boldsymbol{x}, \boldsymbol{o})}{\sum_{\boldsymbol{o}} \sum_{\boldsymbol{c}^+} p_{\text{user}}(\boldsymbol{c}^+|\boldsymbol{x}) \cdot \pi_{\theta_t}(\boldsymbol{o}|\boldsymbol{x}, \boldsymbol{c}^+) \cdot p_{\text{env}}(\boldsymbol{c}|\boldsymbol{x}, \boldsymbol{o})}. \quad (4)$$

The optimization objective for updating $\theta_{t+1}$ based on $\theta_t$ (with gradients stopped through $\theta_t$) is

$$\min_{\pi_{\theta_{t+1}}} \mathbb{E}_{P_{\theta_t}(\boldsymbol{c}|\boldsymbol{x})} \left[ \mathbb{D}_{\text{KL}}(P_{\theta_t}(\boldsymbol{o}|\boldsymbol{x}, \boldsymbol{c}) || \pi_{\theta_{t+1}}(\boldsymbol{o}|\boldsymbol{x}, \boldsymbol{c})) \right]$$

$$\Leftrightarrow \max_{\pi_{\theta_{t+1}}} \mathbb{E}_{P_{\theta_t}(\boldsymbol{c}|\boldsymbol{x})} \left[ \mathbb{E}_{P_{\theta_t}(\boldsymbol{o}|\boldsymbol{x}, \boldsymbol{c})} \left[ \log \pi_{\theta_{t+1}}(\boldsymbol{o}|\boldsymbol{x}, \boldsymbol{c}) \right] \right] \quad (5)$$

$$\Leftrightarrow \max_{\pi_{\theta_{t+1}}} \mathbb{E}_{p_{\text{user}}(\boldsymbol{c}^+|\boldsymbol{x})} \left[ \mathbb{E}_{\pi_{\theta_t}(\boldsymbol{o}|\boldsymbol{x}, \boldsymbol{c}^+)} \left[ \mathbb{E}_{p_{\text{env}}(\boldsymbol{c}|\boldsymbol{x}, \boldsymbol{o})} \left[ \log \pi_{\theta_{t+1}}(\boldsymbol{o}|\boldsymbol{x}, \boldsymbol{c}) \right] \right] \right].$$

**Intuition.** In each training round $t$ (distinct from the $s$-th gradient steps taken within a round), the current model $\pi_{\theta_t}$ is conditioned on $\boldsymbol{c}^+$ to sample candidate positive responses. These responses are

---

**Algorithm 2** Online training: Bootstrapping by conditioning on positive feedback (Section 2.2)

---

**Inputs:** Initialize $\theta_0 = \theta_{\text{off}}$ from Algorithm 1, user-desired feedback $p_{\text{user}}(c^+|x)$, environment $p_{\text{env}}(c|x, o)$, instruct. corpus $\mathcal{X}$, training rounds $T$, steps per round $S$, batch size $B$, optimizer $\mathcal{O}$
**Outputs:** The online-bootstrapped parameters $\theta_T$
1: **for** $t = 1$ **to** $T$ **do**
2:     $\theta_t \leftarrow \theta_{t-1}$
3:     **for all** instructions $x \sim \mathcal{X}$ sampled in this round **do**
4:         **Rollout** $\boxed{c^+ \sim p_{\text{user}}(\cdot|x)}$, $\boxed{o \sim \pi_{\theta_{t-1}}(\cdot|x, c^+)}$ then obtain fresh critique $c \sim p_{\text{env}}(\cdot|x, o)$
5:         **Push** $(x, o, c)$ to buffer $\mathcal{B}_{\text{on}}^t$     // $c$ is usually different (at least linguistically) from $c^+$
6:         **Objective:** $\boxed{\max_{\theta_t} \mathbb{E}_{(x,o,c) \sim \mathcal{B}_{\text{on}}^t} [\log \pi_{\theta_t}(o|x, c)]}$ // Taking expectation over $x \sim \mathcal{X}$ in Eq. (5)
7:     **for** $s = 1$ **to** $S$ **do**
8:         Sample $\{(x_i, o_i, c_i)\}_{i=1}^{B} \sim \mathcal{B}_{\text{on}}^t$;    $\theta_t \leftarrow \mathcal{O}.\text{step}\big(\theta_t, \nabla_{\theta_t} \frac{1}{B} \sum_{i=1}^{B} \log \pi_{\theta_t}(o_i|x_i, c_i)\big)$
9: **return** $\theta_T$

---

then re-annotated with fresh feedback $c$ from the environment. Over successive rounds, the model learns to identify cases where conditioning on $c^+$ does not in fact yield positive critiques, while reinforcing those that align with the expected feedback. This iterative process bootstraps the model, progressively strengthening alignment with user-specified positive feedback. Moreover, following Lanchantin et al. (2025), the number of gradient steps $S$ between rounds can be flexibly adjusted, allowing the procedure to interpolate between fully online and semi-online training.

**Test-time usage.** At test time, FCP does not require any environment-generated feedback. If the user wishes, they may provide a desired feedback condition $c_{\text{user}}$. When no such condition is given, we automatically obtain a positive feedback description by prompting an LLM with instructions such as: "`Provide several possible positive feedback descriptions for the following query.`" This produces a suitable $c^+$ for conditioning. The model then generates an output $o$ using the feedback conditional policy $\pi_\theta(o|x, c^+)$. This is a single-pass generation process that does not require any iterative refinement. In practice, inference simply consists of prepending the selected feedback description to the input prompt.

## 3 EXPERIMENTS

We evaluate FCP on mathematical and general reasoning tasks, aiming for a direct comparison with scalar-based methods. We choose reasoning tasks as the testbed because scalarized RL has been especially successful in this domain (Guo et al., 2025; Ma et al., 2025), making it a strong and convincing benchmark. Showing that FCP performs comparably under such demanding conditions provides a rigorous test of its effectiveness. As shown in Section 3.2, FCP indeed matches scalar pipelines, with more design choices presented in our ablation studies (Section 4).

### 3.1 SETUP

**Datasets and models.** For mathematical reasoning, we use `Big-Math` (Albalak et al., 2025), a 251k-problem dataset curated for training and evaluation. For general reasoning, we use `WebInstruct` (Yue et al., 2024) from GENERAL-REASONER (Ma et al., 2025). Its multi-domain, free-form answers are unsuitable for rule-based filters, so prior work relies on *generative reward model*—making it a natural testbed to contrast verbal conditioning with scalar-reward pipelines. As pilot experiments, our base model is `Qwen2.5-7B-base` (Yang et al., 2024).

**Feedback environment simulation.** Human feedback is costly and difficult to standardize in both quality and style. We therefore simulate the feedback environment with `GPT-5-nano`, which provides feedback for both offline (Algorithm 1) and online (Algorithm 2) training. Our method only requires feedback to be *non-deceptive* (following $p_{\text{env}}$), rather than a detailed breakdown, making lightweight models sufficient. To implement this, we design a unified prompt template (Figure 6) that first elicits a low-quality, *real-world user*-style feedback, then a high-quality, *professional reviewer*-style feedback covering multiple aspects, and finally a *scalar score* summarizing overall quality. This setup ensures that the same feedback source supplies both the *verbal conditions* for FCP and the *scalar rewards* for RL baselines, enabling a fair comparison.

Table 1: **Math (in-domain) and IFEval (out-of-distribution) results.** Here **Avg.** denotes mean accuracy (%) over five math benchmarks. CFT is critique finetuning (Wang et al., 2025a), see Section 4.2, and Critique-GRPO is adopted from Zhang et al. (2025).

| Offline Algo. + Online Algo. | Scalar Reward | In-Domain | | | | | | OOD IFEval |
|---|---|---|---|---|---|---|---|---|
| | | AIME24 | AIME25 | MATH500 | Minerva | Olympiad | Avg. | |
| Base | - | 7.5 ±1.7 | 6.7 ±0.0 | 63.8 ±63.8 | 28.3 ±0.8 | 28.6 ±0.4 | 27.0 ±0.5 | 36.8 |
| + GRPO | ✓ | 20.0 ±0.0 | **13.3** ±2.7 | 75.7 ±1.7 | 42.3 ±1.3 | **40.8** ±0.5 | 38.4 ±0.9 | 38.5 |
| + Critique-GRPO | ✓ | 15.0 ±1.9 | 9.2 ±1.7 | **76.8** ±0.3 | 36.1 ±0.9 | 40.1 ±0.5 | 35.4 ±0.6 | **39.0** |
| RFT | ✓ | 13.3 ±0.0 | 3.3 ±0.0 | 69.2 ±0.6 | 32.4 ±0.6 | 33.8 ±0.9 | 30.4 ±0.0 | 37.5 |
| + GRPO | ✓ | **25.8** ±1.7 | 9.2 ±1.7 | 75.1 ±0.7 | 36.8 ±0.9 | 38.9 ±0.1 | 37.1 ±0.5 | 38.8 |
| + Critique-GRPO | ✓ | 16.7 ±4.7 | 9.2 ±5.0 | 75.2 ±0.4 | 35.8 ±0.7 | 39.6 ±0.5 | 35.3 ±1.2 | 38.6 |
| CFT | ✗ | 1.7 ±3.3 | 0.0 ±0.0 | 27.0 ±3.2 | 9.2 ±6.4 | 7.7 ±1.4 | 9.1 ±1.3 | - |
| **FCP** | ✗ | 6.7 ±0.0 | 3.3 ±3.8 | 68.9 ±1.0 | 31.2 ±1.1 | 32.4 ±1.1 | 28.5 ±1.1 | 38.6 |
| + Bootstrap | ✗ | 25.0 ±3.3 | 7.5 ±1.7 | 76.5 ±0.7 | **45.8** ±0.7 | 38.8 ±0.6 | **38.7** ±0.7 | **39.0** |

**Baselines.** We compare against two strong baselines: Rejection Sampling Finetuning (RFT) and GRPO (Dong et al., 2023; Shao et al., 2024). RFT filters responses by correctness and finetunes only on the correct ones, which in the *offline* case reduces to training on a binary scalar score (correct/incorrect). While simple and effective, it depends on reliable filtering and a stable verifier. GRPO instead uses group-normalized scalar rewards to estimate advantages and has become one of the strongest *online* methods, especially in math reasoning where answers can usually be verified automatically. Both baselines rely on scalar-based filtering or scoring, making them dependent on high-quality verifiable data and an auxiliary verifier. Even rubric-based reward shaping (Zhou et al., 2025b) still loses much of the feedback richness. Our experiments thus offer a stringent comparison between scalar-reward pipelines (RFT/GRPO) and FCP learning.

**Training details for FCP.** In the *offline* stage (Algorithm 1), the base model generates 8 candidate responses per prompt. We discard prompts where all responses are entirely correct or incorrect, then sample one correct and one incorrect response for `GPT-5-nano` to provide feedback. All collected feedback is used to train FCP, while a pool of positive feedback $\{c^+\}$ is built from the scalar scores in the feedback. In the *online* stage (Algorithm 2), for each prompt $x$ we sample a desired condition $c^+ \sim p_{\text{user}}(\cdot|x)$ by drawing from the pool $\{c^+\}$. For rollout, the prompt batch size is 2048 with 4 responses per prompt; for training, the mini-batch size is 512, giving 4 gradient updates per rollout step. Each response receives a fresh *professional reviewer*-style feedback from `GPT-5-nano`, which is concatenated with the prompt and response (using the Algorithm 1 wrapper `<EF>` and `</EF>`) for cross-entropy training. This bootstrapping loop improves response quality under desired conditions while grounding updates in new feedback. For fair comparison, GRPO is trained with the same scalar scores from `GPT-5-nano` under the identical prompt template.

**Evaluation.** We assess mathematical reasoning on AIME24&25, MATH500 (Hendrycks et al., 2021), Minerva-Math (Lewkowycz et al., 2022), and OlympiadBench (He et al., 2024), and general reasoning on GPQA-Diamond (Rein et al., 2024), MMLU-Pro (Wang et al., 2024), and TheoremQA (Chen et al., 2023). To test instruction-following beyond the training domain, we also include IFEval (Zhou et al., 2023). All benchmarks use a unified protocol: each dataset is run under four random seeds [2], with mean accuracy reported. Inference uses `vllm` (Kwon et al., 2023) with greedy decoding and a maximum generation length of 8192 tokens. For FCP, we match the training setup by randomly sampling one feedback condition from $\{c^+\}$ for each question and prepending it to the prompt template.

## 3.2 MAIN RESULTS

**Offline FCP is comparable to RFT.** On `Qwen2.5-7B-base`, offline FCP attains 28.8% average accuracy on the math suite, between the base model (27.0%) and RFT (30.4%) (Table 1). General reasoning shows the same order: 38.7%, 43.5%, and 44.6% for base, FCP, and RFT (Table 2). This is expected, since FCP directly learns from all response-feedback pairs without filtering and therefore inevitably absorbs noise, whereas RFT benefits from elaborate correctness filtering. Still, FCP remains competitive under noisier supervision.

---

[2]For MMLU-Pro, the large test set size (~12k questions) simultaneously leads to a high evaluation cost and an inherently small variance (typically ≤ 0.3). Consequently, we report results based on a single evaluation run for this benchmark.

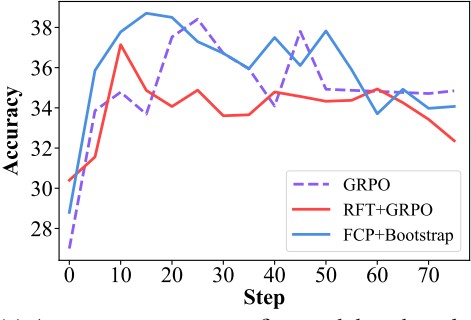 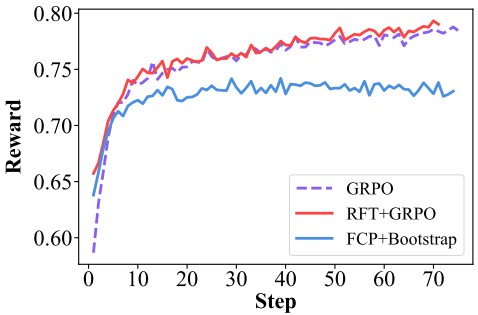

(a) Average accuracy over five math benchmarks measured at intermediate checkpoints.

(b) Scalar scores assigned by `GPT-5-nano` to model rollouts during training.

Figure 2: **Training dynamics of FCP and scalar-based baselines.** (a) FCP+Bootstrap matches GRPO and RFT+GRPO accuracy within 30 steps. (b) In contrast, its scalar reward scores lag behind, consistent with the fact that FCP does not directly optimize against reward model's preference.

Table 2: **General reasoning results.** Accuracy (%) across three benchmarks and their average.

| Offline Algo. + Online Algo. | Scalar Reward | GPQA-Diamond | MMLU-Pro | TheoremQA | Average |
|---|---|---|---|---|---|
| Base | - | 27.9 $_{\pm 1.0}$ | 49.7 | 38.6 $_{\pm 0.2}$ | 38.7 $_{\pm 0.4}$ |
| + GRPO | ✓ | 32.5 $_{\pm 5.3}$ | 49.7 | **49.4** $_{\pm 1.4}$ | 43.9 $_{\pm 1.7}$ |
| RFT | ✓ | 35.2 $_{\pm 1.3}$ | 55.0 | 43.7 $_{\pm 0.9}$ | 44.6 $_{\pm 0.2}$ |
| + GRPO | ✓ | 37.2 $_{\pm 2.5}$ | **57.0** | 48.3 $_{\pm 0.2}$ | 47.5 $_{\pm 0.8}$ |
| **FCP** | ✗ | 35.0 $_{\pm 2.9}$ | 53.6 | 42.0 $_{\pm 1.0}$ | 43.5 $_{\pm 0.6}$ |
| **+ Bootstrap** | ✗ | **39.1** $_{\pm 2.9}$ | 55.3 | 49.1 $_{\pm 0.5}$ | **47.8** $_{\pm 0.9}$ |

**Bootstrapping enables FCP to rival scalarized RL baselines.** Online bootstrapping lifts FCP from 28.8% to 38.7% average accuracy on the math suite (Table 1), slightly surpassing GRPO (38.4%). A similar trend appears in out-of-distribution case: on **IFEval**, FCP+Bootstrap reaches 39.0%, comparable to GRPO (38.5%) and RFT+GRPO (38.8%). General reasoning benchmarks (Table 2) show the same pattern, with FCP+Bootstrap at 47.8%, matching the best scalar-based baseline (47.5%). These results indicate that bootstrapping gives FCP the effectiveness of scalarized RL while retaining the advantage of learning directly from richer verbal feedback.

### 3.3 LEARNING DYNAMICS OF FCP

**FCP enables controllable behavior across diverse feedback conditions.** A core question is whether the policy truly *learns* the conditioning signal $c$—and, if so, whether this lets us absorb negative samples into training without hurting best-case performance. We probe this by sampling representative feedback from the offline pool and evaluating under several conditions.

Table 3 shows a sharp contrast on MATH500: accuracy is 68.5% under `fully_positive` but only 17.1% under `fully_negative`, far below the base model's 63.8% (Table 1). This indicates the model internalizes the control signal: negative conditions induce poor behavior when requested, yet positive conditions still yield strong accuracy—showing that including negative samples in training (*using the same cross-entropy loss as positives*) does not cap performance under positive ones.

Other conditions also shift behavior as intended. Under `neutral`, where the condition $c$ asks for a correct answer and a more verbose solution, accuracy drops slightly but response length grows, reflecting a trade-off. With `has_code`, the share of responses containing code rises to 74.3%, confirming that stylistic attributes in $c$ are also followed. Compared to `Qwen2.5-7B-Instruct`, which shows little variation across conditions due to training only on verified positives, FCP learns to map feedback $c$ to distinct behaviors, enabling broad data use without manual filtering.

**FCP achieves strong accuracy without over-optimizing scalar rewards.** As seen in Figure 2a, both FCP and GRPO reach peak accuracy within 30 online steps, with scalar scores from `GPT-5-nano` rising sharply at the start. Yet Figure 2b shows FCP's scores lagging behind GRPO's later, since it does not directly optimize against the scalar reward model. Crucially, FCP sustains high accuracy

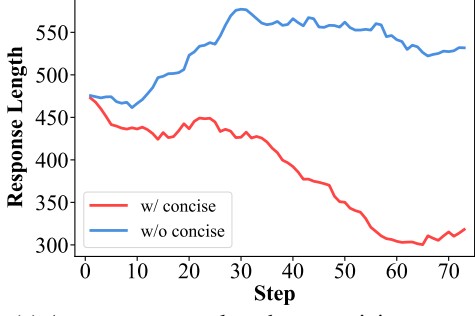 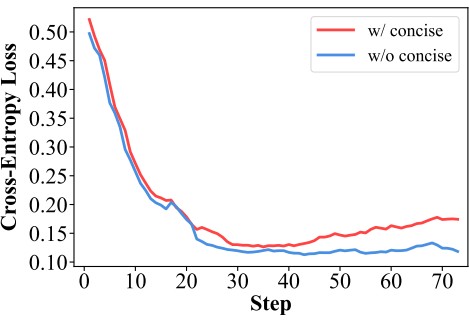

(a) Average response length over training steps. (b) Cross-entropy loss over training steps.

Figure 3: **Effect of length-related conditions on bootstrapping stability.** Both curves are smoothed with a 10-step moving average. (a) Without filtering, response length decreases over time, while filtering out length-related conditions leads to steady growth. (b) The corresponding loss curves show greater instability when length-related conditions are included.

Table 3: **Comparison under different feedback conditions.** Accuracy (%), code ratio (proportion of responses containing code), and average response length are all measured on MATH500.[3]

| | | Example 1 | Example 2 | Example 3 | Example 4 |
|---|---|---|---|---|---|
| **Feedback type** | | `fully_positive` | `fully_negative` | `neutral` | `has_code` |
| **Content** | | Accurate and clear; concise and coherent reasoning; correct conclusion. | Incoherent and incomplete. Random and unfocused. Unclear and disorganized. | Correct and readable overall, but the solution is **verbose** and could be streamlined for tighter logical flow. | Correct and clear, though slightly verbose with superfluous **code**. |
| **Accuracy** | Instruct | 76.2 | 77.4 | 77.5 | 76.6 |
| | FCP | 68.5 | **17.1** | 61.1 | 53.9 |
| **Code Ratio** | Instruct | 0 | 0 | 0 | 0 |
| | FCP | 22.7 | 55.6 | 46.3 | **74.3** |
| **Response Length** | Instruct | 632 | 650 | 638 | 661 |
| | FCP | 605 | 1442 | **722** | 659 |

despite lower scores, indicating it avoids the reward-hacking behavior often seen in scalar-based methods and underscoring verbal feedback as a more robust training signal.

**Length-related conditions destabilize FCP bootstrapping.** We find that feedback conditions $c^+$ tied to output length, such as *conciseness*, can destabilize online bootstrapping. As shown in Figure 3, these conditions cause average response length to shrink over time while the loss becomes unstable. This likely reflects a feedback loop: concise rollouts receive affirming feedback, and cross-entropy updates further shorten responses, eventually collapsing output length. Filtering out length-related conditions instead yields steadily longer responses, mirroring GRPO's training behavior (Guo et al., 2025) and supporting the view that reliable math solving benefits from extended reasoning traces.

## 4 ABLATION STUDIES

Unless otherwise noted, we use the following *default* configuration: For rollout, the prompt batch size is 512 with 4 responses generated per prompt. For training, the mini-batch size is 512, corresponding to a single gradient update per rollout step, which yields a fully online setting. All rollouts of the same prompt share an identical feedback condition $c^+$. Training uses token-level mean loss aggregation, with fresh feedback $c$ provided in the professional *reviewer*-style by GPT-5-nano.

### 4.1 REAL-WORLD USER VS. PROFESSIONAL REVIEWER STYLE

Real-world user feedback is abundant and inexpensive but often noisy and inconsistent; professional reviewer feedback is higher quality but costly and less scalable. We therefore ask: *how much feedback quality does FCP actually require?* We use a unified prompt that asks GPT-5-nano to produce both a low-quality real-world *user*-style feedback and a high-quality professional *reviewer*-style feedback in a single response. As shown in Table 4, *user*-style feedback is typically subjective and colloquial, whereas *reviewer*-style feedback is precise and structured.

---

[3]For the Instruct model, evaluation prompts are wrapped as "`Your answer should be expected to get the following critique: <feedback_content>\n{question}`".

Table 4: Examples of feedback in *real-world user*-style and *professional reviewer*-style.

| Role | Critique Type | Examples |
|------|---------------|----------|
| Real-World User | `fully_positive` | That looks right to me, concise and easy to follow. I'm satisfied with the final result. |
| | `fully_negative` | I have no idea what you were trying to say—the response is nonsense and not helpful at all. |
| | `neutral` | I'm not completely sure about the logic, but the final answer matches the number I was expecting. |
| Professional Reviewer | `fully_positive` | Correct and clear; succinct and logically sound, with concise and effective reasoning. |
| | `fully_negative` | Incorrectly structured and incoherent. The reasoning is absent and the content is unusable. |
| | `neutral` | Correct final result but unclear and incomplete reasoning; concise yet insufficiently rigorous. |

Table 5: Ablation results on hyperparameter choices, data sources, and feedback settings. Reported numbers are average accuracy on math benchmarks; $\Delta$ shows change relative to the default setting.

| Variant | Changed Setting(s) | Avg Acc | $\Delta$ |
|---------|--------------------|---------|----------|
| **Default** | —— | 35.3 | 0.0 |
| w/ user style feedback | `critique_type=user` | 32.8 | -2.5 |
| w/ partial online | `prompt_bsz=2048` | 38.7 | +3.4 |
| w/ unbiased loss | `loss_agg_mode=seq-mean-token-sum` | 36.0 | +0.7 |
| w/ smaller batch size | `train_bsz=ppo_mini_bsz=256` | 37.7 | +2.4 |
| w/ more diverse $c^+$ | use random $c^+$ per rollout | 34.1 | -1.2 |
| w/ different dataset | use MATH-Train split | 34.3 | -1.0 |

Table 5 shows that using only *user*-style feedback (offline and online) lowers math-suite accuracy by 2.5 points relative to *reviewer*-style feedback, yet still delivers a +5.8 gain over the base model (27%; Table 1). While *reviewer*-style feedback is more effective, *user*-style feedback remains surprisingly competitive after FCP training. Its lower cost and broad availability make it a practical source for scaling, with *reviewer*-style feedback reserved for targeted quality improvements.

## 4.2 ADDITIONAL TRAINING DESIGN CHOICES AND COMPARISON TO CFT

We further study how different design choices affect FCP training, with results summarized in Table 5.

**Online update strategy.** Compared to the fully online setup, using a larger prompt batch size of 2048 while keeping the mini-batch size fixed at 512 results in four gradient updates per rollout step, and yields better accuracy. This suggests that partial online updates can improve optimization efficiency.

**Loss aggregation.** In Algorithm 2, cross-entropy on self-sampled responses reduces to policy gradient with unit advantages, which suffers from length bias (Liu et al., 2025a). A debiased scheme averaging at the sequence level and summing at the token level gives a consistent +0.7% gain.

**Other variations.** We also experimented with several alternative configurations. Reducing the training batch size to 256 improves accuracy by about +2.4%. Training the online stage on a dataset different from that used for offline pretraining slightly underperforms the default baseline, yet remains +5.5% above the offline-only initialization, indicating that offline and online datasets need not be strictly aligned for FCP to be effective.

**Comparison to Critique Finetuning (CFT).** CFT can perform well with high-quality and detailed critiques (Wang et al., 2025a), but applying it to the same coarse and lightweight feedback used for FCP leads to severe degradation—worse than the base model (Table 1). This highlights a key strength of FCP: it effectively leverages coarse, high-level feedback without costly fine-grained annotations.

## 5 RELATED WORK

**SFT and RL methods for reasoning.** The ability to perform reasoning has become a defining strength of LLMs, enabling progress across mathematics, coding, and scientific domains (Jaech et al., 2024; Comanici et al., 2025). To enhance these skills, two approaches have proven especially influential: SFT and RL (Uesato et al., 2022; Rafailov et al., 2023; Guha et al., 2025; Hu et al.,

2025; Hochlehnert et al., 2025). Following the success of the DeepSeek-R1 recipe (Shao et al., 2024; Guo et al., 2025), a number of RL variants have been introduced, including Dr. GRPO (Liu et al., 2025a), DAPO (Yu et al., 2025), REINFORCE++ (Hu, 2025), and VAPO (Yue et al., 2025). Beyond algorithmic proposals, researchers have systematically investigated the RL design space for reasoning (Zeng et al., 2025; Team et al., 2025), examining factors such as staged training curricula (Wen et al., 2025; Luo et al., 2025) and reward formulation (Gao et al., 2024; Cui et al., 2025; Qi et al., 2025; Zhou et al., 2025a). While much of the initial progress focused on mathematics, these methods have more recently been extended to software engineering and code reasoning (Liu & Zhang, 2025; Xie et al., 2025; Wei et al., 2025; Yang et al., 2025; Chen et al., 2025), as well as to broader agentic applications (Wang et al., 2025b; Jin et al., 2025; Jiang et al., 2025; Xue et al., 2025).

**Learning from verbal feedback.** Most existing approaches convert verbal feedback into scalar rewards for RL training (Kim et al., 2024; Ankner et al., 2024; Lightman et al., 2024; Stephan et al., 2024; Whitehouse et al., 2025; Liu et al., 2025b). More recent efforts explore learning directly from feedback or critiques: Lloret et al. (2024) propose conditional SFT based on toxicity categorization in alignment tasks, CFT (Wang et al., 2025a) trains models to imitate critiques, Critique-GRPO (Zhang et al., 2025) incorporates critique-guided refinements into online RL, Salemi & Zamani (2025) jointly optimize a feedback model and a policy model, and Chen et al. (2024) introduce a refinement model that corrects errors using feedback. These approaches generally assume feedback is high-quality, informative or categorized, and reliably improves self-refinement. In practice, however, human feedback is often mixed, free-form, emotional, or uncertain. Moreover, while such feedback is easy to collect, its distribution is difficult to model with generative reward models that must capture diverse user interaction styles. In contrast, our FCP framework does not require feedback to be high-quality or rubric-constrained; by treating feedback as a conditioning signal rather than a prediction target, it can flexibly exploit the full range of verbal feedback, including noisy or mixed forms, for training.

## 6 DISCUSSION AND FUTURE DIRECTIONS

Our key insight is that the essence of RL lies in *online interaction with the environment*, not in scalar rewards or any specific algorithm. Scalarization was historically necessary for control-centric RL in robotics or strategy-centric RL in games, but it may not be intrinsic to language-centric systems like LLMs. This reopens the debate around the reward hypothesis: earlier critics could only offer counterexamples without an alternative framework (Skalse & Abate, 2022), whereas our FCP approach leverages **language priors** to provide a principled way to bypass scalar rewards. Crucially, during training, feedback $c$ is a *dependent variable* generated from the environment $p_{env}(c|x, o)$ and cannot be directly controlled, while at test time the conditioning feedback $c^+$ becomes an *independent variable* freely specified by users. This asymmetry enables full use of diverse feedback during training while allowing precise controllability at inference. By directly mapping feedback to responses, our FCP bypasses reward imbalance, preserves feedback richness, and improves data efficiency. Unlike RFT (Dong et al., 2023; Touvron et al., 2023), which discards many useful data pairs, FCP retains diverse feedback, including mixed and uncertain, and can merge complementary signals across examples at test time (Figure 1). This establishes verbal feedback as a first-class training signal and FCP as a natural, scalable alternative to scalarized RL.

**Future directions.** Several extensions of FCP are promising. One is to *combine it with verifiable rewards*, for instance by treating the absence of feedback as a neutral condition (e.g., using the null feedback token `<EF></EF>`), so that reliable scalar supervision can complement verbal feedback when available. Another is to extend FCP to *multi-turn interactions*, where feedback is incorporated before the next turn of generation in a teacher-forcing style, enabling closer alignment with iterative human guidance. A third is *test-time adaptation*: by conditioning on a few user-provided examples, the model could rapidly adjust to individual feedback styles, similar to personalization in text-to-image generation. Finally, the feedback condition $c$ could be made *multimodal*. Collectively, these future directions would deepen integration of natural feedback into LLM training, bridging offline and online stages while adapting to diverse user needs.

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

**Training phase**: Collecting images from $P_{\text{Image}}$ and captions from $P_{\text{Caption}}$ | **Test phase**: Users generate novel images from $P_{\text{T2I}}$

$\mathcal{I}_1 \sim P_{\text{Image}}(\cdot)$ $\qquad$ $\mathcal{I}_2 \sim P_{\text{Image}}(\cdot)$

A banana and an apple on the table $\qquad$ A man is surfing on an ocean wave

$\mathcal{T}_1 = (\mathcal{T}_1^+, \mathcal{T}_1^-) \sim P_{\text{Caption}}(\cdot|\mathcal{I}_1)$ $\quad$ $\mathcal{T}_2 = (\mathcal{T}_2^+, \mathcal{T}_2^-) \sim P_{\text{Caption}}(\cdot|\mathcal{I}_2)$

A banana is surfing on an ocean wave

$\mathcal{T}^+ = (\mathcal{T}_1^+, \mathcal{T}_2^+)$ is expected by users

$\mathcal{I}^+ \sim P_{\text{T2I}}(\cdot|\mathcal{T}^+)$

Figure 4: **Learning from mixed captions in text-to-image generation.** During training, models learn from realistic image-caption pairs such as "*a banana and an apple on the table*" or "*a man surfing on an ocean wave*". They can leverage language priors to recombine these captions and generate novel concepts, such as "*a banana surfing on the ocean*" (images shown are generated with Gemini 2.5 Flash Image). By analogy to Figure 1, this illustrates how diverse verbal feedback can be treated as a conditioning signal, motivating our feedback-conditional learning paradigm.

# A ADDITIONAL DERIVATIONS AND DISCUSSIONS

## A.1 PROOF OF EQ. (2) AND ITS SPECIAL CASE

Following Rafailov et al. (2023), the optimal solution to a KL-constrained reward maximization problem $\mathbb{E}_{\pi(o|x,c^+)}\left[\log p_{\text{env}}(c^+|x,o)\right] - \mathbb{D}_{\text{KL}}\left(\pi(o|x,c^+)||\pi_{\text{ref}}(o|x)\right)$ can be written as

$$
\begin{aligned}
\pi^*(o|x,c^+) &= \frac{\pi_{\text{ref}}(o|x) \cdot \exp\left(\log p_{\text{env}}(c^+|x,o)\right)}{\sum_o \pi_{\text{ref}}(o|x) \cdot \exp\left(\log p_{\text{env}}(c^+|x,o)\right)} \\
&= \frac{\pi_{\text{ref}}(o|x) \cdot p_{\text{env}}(c^+|x,o)}{\sum_o \pi_{\text{ref}}(o|x) \cdot p_{\text{env}}(c^+|x,o)} \\
&= \frac{P_{\text{off}}(o,c^+|x)}{P_{\text{off}}(c^+|x)} = P_{\text{off}}(o|x,c^+).
\end{aligned}
\tag{6}
$$

Note that the objective in Eq. (2) is equivalent to minimizing the *reverse* KL divergence between $\pi(o|x,c^+)$ and $P_{\text{off}}(o|x,c^+)$:

$$
\begin{aligned}
&\mathbb{E}_{\pi(o|x,c^+)}\left[\log p_{\text{env}}(c^+|x,o)\right] - \mathbb{D}_{\text{KL}}\left(\pi(o|x,c^+)||\pi_{\text{ref}}(o|x)\right) \\
&= -\mathbb{D}_{\text{KL}}\left(\pi(o|x,c^+)||P_{\text{off}}(o|x,c^+)\right) + \log P_{\text{off}}(c^+|x).
\end{aligned}
\tag{7}
$$

In the special case where the environment provides *verifiable rewards*, that is, $p_{\text{env}}(c^+|x,o^+) = 1$ for correct responses $o^+$ and $p_{\text{env}}(c^+|x,o^-) = 0$ for incorrect responses $o^-$, we can show that $P_{\text{off}}(o|x,c^+)$ reduces to the optimal solution of a 0-1 reward maximization problem without KL regularization: $P_{\text{off}}(o|x,c^+) \in \arg\max_\pi \mathbb{E}_{\pi(o|x,c^+)}\left[\mathbb{1}(o \text{ is } o^+)\right]$. Specially, we have

$$
\begin{aligned}
P_{\text{off}}(o^+|x,c^+) &= \frac{\pi_{\text{ref}}(o^+|x) \cdot p_{\text{env}}(c^+|x,o^+)}{\sum_o \pi_{\text{ref}}(o|x) \cdot p_{\text{env}}(c^+|x,o)} = \frac{\pi_{\text{ref}}(o^+|x)}{\sum_{o \text{ is } o^+} \pi_{\text{ref}}(o|x)}; \\
P_{\text{off}}(o^-|x,c^+) &= \frac{\pi_{\text{ref}}(o^-|x) \cdot p_{\text{env}}(c^+|x,o^-)}{\sum_o \pi_{\text{ref}}(o|x) \cdot p_{\text{env}}(c^+|x,o)} = 0.
\end{aligned}
\tag{8}
$$

Thus, taking $\pi(o|x,c^+) = P_{\text{off}}(o|x,c^+)$ into the formula of $\mathbb{E}_{\pi(o|x,c^+)}\left[\mathbb{1}(o \text{ is } o^+)\right]$, we have

$$
\mathbb{E}_{P_{\text{off}}(o|x,c^+)}\left[\mathbb{1}(o \text{ is } o^+)\right] = \sum_{o \text{ is } o^+} \frac{\pi_{\text{ref}}(o|x) \cdot \mathbb{1}(o \text{ is } o^+)}{\sum_{o \text{ is } o^+} \pi_{\text{ref}}(o|x)} = 1.
\tag{9}
$$

Since there is $\max_\pi \mathbb{E}_{\pi(o|x,c^+)}\left[\mathbb{1}(o \text{ is } o^+)\right] = 1$, we know that $\pi(o|x,c^+) = P_{\text{off}}(o|x,c^+)$ is one of the optimal solutions (not unique), i.e., $P_{\text{off}}(o|x,c^+) \in \arg\max_\pi \mathbb{E}_{\pi(o|x,c^+)}\left[\mathbb{1}(o \text{ is } o^+)\right]$. $\qquad\square$

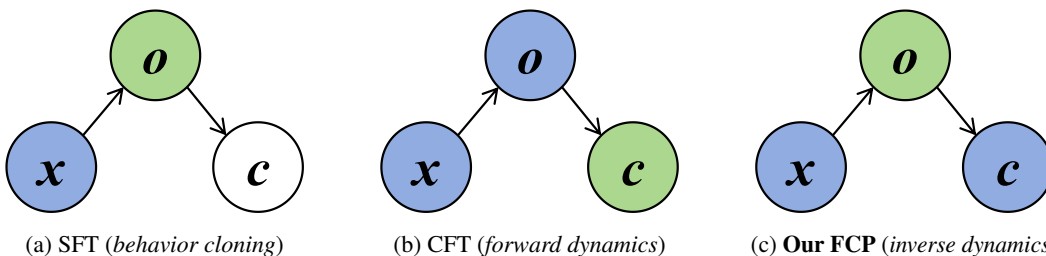

(a) SFT (*behavior cloning*)    (b) CFT (*forward dynamics*)    (c) **Our FCP** (*inverse dynamics*)

Figure 5: **Graphical models for SFT, CFT, and our FCP.** Following Brandfonbrener et al. (2023), we use blue color to indicate inputs to the algorithm and green color to indicate prediction targets.

## A.2    CONNECTION TO INVERSE DYNAMICS MODELING

In traditional RL, objectives for representation learning are often grouped into three classes: **behavior cloning**, **forward dynamics**, and **inverse dynamics**. Behavior cloning is typically used for imitation learning (Arora et al., 2020; Zang et al., 2022), forward dynamics is central to world modeling (Ha & Schmidhuber, 2018; Schwarzer et al., 2021), and inverse dynamics has been explored for both pretraining (Brandfonbrener et al., 2023) and feature extraction for exploration in RL (Efroni et al., 2021).

Interestingly, analogous structures appear in the LLM literature. The objectives of supervised finetuning (SFT), critique finetuning (CFT) (Wang et al., 2025a), and our feedback-conditional policy (FCP) align naturally with behavior cloning, forward dynamics, and inverse dynamics, respectively:

$$\textbf{SFT (behavior cloning):} \quad \max_{\pi_\theta} \mathbb{E}_{\pi_{\text{ref}}(\boldsymbol{o}|\boldsymbol{x})} \left[ \log \pi_\theta(\boldsymbol{o}|\boldsymbol{x}) \right];$$

$$\textbf{CFT (forward dynamics):} \quad \max_{\pi_\theta} \mathbb{E}_{\pi_{\text{ref}}(\boldsymbol{o}|\boldsymbol{x})} \left[ \mathbb{E}_{p_{\text{env}}(\boldsymbol{c}|\boldsymbol{x},\boldsymbol{o})} \left[ \log \pi_\theta(\boldsymbol{c}|\boldsymbol{x},\boldsymbol{o}) \right] \right]; \quad (10)$$

$$\textbf{Our FCP (inverse dynamics):} \quad \max_{\pi_\theta} \mathbb{E}_{\pi_{\text{ref}}(\boldsymbol{o}|\boldsymbol{x})} \left[ \mathbb{E}_{p_{\text{env}}(\boldsymbol{c}|\boldsymbol{x},\boldsymbol{o})} \left[ \log \pi_\theta(\boldsymbol{o}|\boldsymbol{x},\boldsymbol{c}) \right] \right].$$

We further illustrate this categorization with graphical models in Figure 5. This unified perspective clarifies the conditional structure underlying each finetuning paradigm and highlights how different forms of supervision drive model learning. In particular, our FCP extends the analogy by treating verbal feedback as a first-class supervision signal, positioning it as the natural *inverse-dynamics* counterpart to existing finetuning objectives.

## B    DETAILED EXPERIMENTAL SETUP

All implementations are based on `llama-factory` (Zheng et al., 2024) and `verl` (Sheng et al., 2025). Hyperparameter settings for both offline and online stages of FCP are listed in Table 6.

For the two special tokens `<EF>` and `</EF>`, embeddings are initialized by sampling from a multivariate normal distribution with mean and covariance computed over existing token embeddings. For general reasoning bootstrapping, we adopt a fully online setup with batch size of 256, differing from the math setting to illustrate that FCP remains effective under both training strategies.

Finally, Figure 6 shows the unified prompt template used to elicit feedback from `GPT-5-nano`. The template produces three outputs in one response: a low-quality *real-world user*-style feedback, a high-quality *professional reviewer*-style feedback, and a scalar score summarizing overall quality.

## C    MORE EXPERIMENT RESULTS

## D    LLM USAGE

We used an OpenAI LLM (GPT-5) as a writing and formatting assistant. In particular, it helped refine grammar and phrasing, improve clarity, and suggest edits to figure/table captions and layout (e.g., column alignment, caption length, placement). The LLM did not contribute to research ideation, experimental design, implementation, data analysis, or technical content beyond surface-level edits. All outputs were reviewed and edited by the authors, who take full responsibility for the final text and visuals.

Table 6: **Hyperparameters for FCP training** used in the offline and bootstrapping (online) stages.

| Hyperparameter | Offline | Online |
|---|---|---|
| learning rate | 5e-6 | 1e-6 |
| lr scheduler | cosine | constant |
| weight decay | 0 | 0.01 |
| warmup ratio | 0.1 | 0 |
| train batch size | 512 | 2048 |
| ppo mini-batch size | — | 512 |
| temperature | — | 1.0 |
| top_p | — | 1.0 |
| rollout_n | — | 4 |
| epoch | 1 | |
| max response length | 4096 | |
| loss type | cross-entropy loss | |
| loss aggregation mode | token-mean | |
| feedback environment | `GPT-5-nano` | |
| feedback style | *professional reviewer* | |

Table 7: Performance comparison under verifiable (rule-based) and LLM-generated supervision across multiple training methods.

| Method | Source | AIME24 | AIME25 | MATH500 | Minerva | Olympiad | Avg. |
|---|---|---|---|---|---|---|---|
| Base | - | 7.5 | 6.7 | 63.8 | 28.3 | 28.6 | 27.0 |
| Base + GRPO | rule-based verifier | 13.3 | 14.2 | 76.3 | 36.6 | 41.7 | 36.4 |
| Base + GRPO | LLM | 20.0 | 13.3 | 75.7 | 42.3 | 40.8 | 38.4 |
| RFT + GRPO | rule-based verifier | 17.5 | 15.0 | 77.0 | 38.4 | 41.3 | 37.8 |
| RFT + GRPO | LLM | 25.8 | 9.2 | 75.1 | 36.8 | 38.9 | 37.1 |
| FCP + Bootstrap | LLM | 25.0 | 7.5 | 76.5 | 45.8 | 38.8 | 38.7 |

Table 8: Evaluation Results Under In-Distribution (ID) and Out-of-Distribution (OOD) Feedback Conditions

| Method | Feedback Style | ID/OOD | AIME24 | AIME25 | MATH500 | Minerva | Olympiad | Avg. |
|---|---|---|---|---|---|---|---|---|
| Base | - | - | 7.5 | 6.7 | 63.8 | 28.3 | 28.6 | 27.0 |
| FCP + Bootstrap | Reviewer | ID | 25.0 | 7.5 | 76.5 | 45.8 | 38.8 | 38.7 |
| FCP + Bootstrap | User | OOD | 10.8 | 7.5 | 75.1 | 35.2 | 38.0 | 33.3 |
| FCP + Bootstrap | "Correct" | OOD | 16.7 | 7.5 | 75.3 | 35.2 | 37.2 | 34.4 |
| FCP + Bootstrap | No Feedback | OOD | 15.8 | 10.0 | 74.4 | 35.6 | 37.6 | 34.7 |

Table 9: Effects of training-used feedback quality on FCP performance across reasoning benchmarks.

| Method | Feedback Style | Quality | AIME24 | AIME25 | MATH500 | Minerva | Olympiad | Avg. |
|---|---|---|---|---|---|---|---|---|
| Base | - | - | 7.5 $_{\pm1.7}$ | 6.7 $_{\pm0.0}$ | 63.8 $_{\pm63.8}$ | 28.3 $_{\pm0.8}$ | 28.6 $_{\pm0.4}$ | 27.0 $_{\pm0.5}$ |
| FCP + Bootstrap | Correctness-Only | very low | 10.0 $_{\pm2.7}$ | 5.0 $_{\pm1.9}$ | 73.4 $_{\pm0.7}$ | 34.3 $_{\pm0.6}$ | 35.3 $_{\pm0.4}$ | 31.6 $_{\pm0.8}$ |
| FCP + Bootstrap | User | low | 16.7 $_{\pm2.7}$ | 0.8 $_{\pm1.7}$ | 72.2 $_{\pm0.4}$ | 37.1 $_{\pm0.4}$ | 37.1 $_{\pm0.8}$ | 32.8 $_{\pm0.8}$ |
| FCP + Bootstrap | Reviewer-Lite | medium | 14.2 $_{\pm4.2}$ | 8.3 $_{\pm1.9}$ | 74.0 $_{\pm0.3}$ | 37.4 $_{\pm1.5}$ | 37.3 $_{\pm0.7}$ | 34.2 $_{\pm1.0}$ |
| FCP + Bootstrap | Reviewer | high | 25.0 $_{\pm3.3}$ | 7.5 $_{\pm1.7}$ | 76.5 $_{\pm0.7}$ | 45.8 $_{\pm0.7}$ | 38.8 $_{\pm0.6}$ | 38.7 $_{\pm0.7}$ |

Table 10: **Comparison of FCP performance using weak critique models.**

| Method / Model | Critique Model | Feedback Style | AIME24 | AIME25 | MATH500 | Minerva | Olympiad | Avg. |
|---|---|---|---|---|---|---|---|---|
| Qwen2.5-1.5B-Instruct | - | - | 0.0 | 0.0 | 55.0 | 20.5 | 20.3 | 19.2 |
| Qwen2.5-7B-Base | - | - | 7.5 | 6.7 | 63.8 | 28.3 | 28.6 | 27.0 |
| w/ FCP + Bootstrap | Qwen2.5-1.5B-Instruct | User | 14.2 | 5.8 | 72.9 | 35.5 | 37.4 | 33.1 |
| w/ FCP + Bootstrap | GPT-5-nano | User | 16.7 | 0.8 | 72.2 | 37.1 | 37.1 | 32.8 |
| w/ FCP + Bootstrap | GPT-5-nano | Reviewer | 25.0 | 7.5 | 76.5 | 45.8 | 38.8 | 38.7 |

You are acting as a real-world human user of an LLM.

Inputs:
**Question:**
\"\"\"
{question}
\"\"\"

**Model Answer:**
\"\"\"
{model_answer}
\"\"\"

**Reference Final Answer (used only for correctness check):**
\"\"\"
{reference_answer}
\"\"\"

**Your tasks:**

**1) Simulate "user feedback" from a normal, real-world user reacting to the Model Answer only.**
  - Length: 1-3 sentences, colloquial tone, first person.
  - Content: purely subjective sentiment (e.g., helpfulness, confidence, confusion, satisfaction).
  - STRICT: Do NOT mention or allude to any symbols, formulas, variable names, or specialized concepts from the Question or the Model Answer. Do NOT quote text from the inputs.

For example:
"I think you are right, but your solution is really long and complicated."
"You are a genius! You have all my respect."
"I am confused. There seems to be a mistake in your solution."
"What are you talking about? You are not answering my question."
etc.

**2) Simulate a professional reviewer evaluating the Model Answer along several dimensions, including but not limited to:**
  • correctness — Compare the Model Answer's final result ONLY against the Reference Final Answer (if provided). Judge whether the end result matches; do not use the reference for any other purpose.
  • logical_rigor — Assess the soundness and gaplessness of reasoning within the Model Answer itself. Do NOT use the Reference Final Answer here.
  • completeness — Judge coverage of required parts and edge cases based on the Question and the Model Answer only. Do NOT use the Reference Final Answer here.
  • clarity — Evaluate organization, readability, and ease of following in the Model Answer. Do NOT use the Reference Final Answer here.

**Then provide a high-level summary (1-3 sentences) with overall judgment and broad observations.**
- STRICT for the high-level summary: Only use adjectives and adverbs to describe the Model Answer and reasoning process. DO NOT mention where it goes wrong and where it can do better.

For example:
"Your final answer is correct, but the solution is too long and complicated. There are also several logical errors in your solution."
"The answer is partially correct. The reasoning is sound but not complete. Also, you are being too verbose."
"The answer is totally wrong. It lacks soundness and is not complete. However, the solution is concise and clear."

Hard constraints:
- Keep all content in English.
- Do not mention anything like "reference" or "python snippet".

Output format:
### User-style feedback: <your 1-3 sentence feedback>
### Analysis along several dimensions: <your 1-3 sentence analysis>
### High-level summary: <your 1-3 sentence summary>
### Score (0-10): <one overall integer score>

Figure 6: Prompt template used to elicit feedback from `GPT-5-nano`, including *real-world user*-style feedback, *professional reviewer*-style feedback, and a scalar score.

