# OpenReview forum: "Language Models Can Learn from Verbal Feedback Without Scalar Rewards"
_ICLR.cc/2026/Conference — Submitted to ICLR 2026_

### Official Review · Reviewer_8GhM · 2025-10-24

**Soundness:** 3
**Presentation:** 3
**Contribution:** 2
**Rating:** 4
**Confidence:** 3

**Summary:**

In classical reinforcement learning approaches,  feedback from the environment are usually represented as scalar rewards. While this principle has guided the design of traditional RL algorithms, it becomes restrictive in the context of language models, where feedback is often expressed in rich verbal form. Training LLMs with such scalar feedback leads has severe limitations and is prone to a significant loss of information.

The authors introduce a feedback-conditional policy (FCP) framework. In the offline stage, a reference policy model receives an input instruction, generates a response, and subsequently obtains verbal feedback from the environment. This enables the definition of a feedback-conditional posterior distribution, the posterior distribution of the outputs given the input and the feedback. Since this distribution is generally intractable, the authors propose to approximate it by learning a feedback-conditional policy via minimization of the forward Kullback–Leibler (KL) divergence. The learned model is thereby capable of generating outputs conditioned on arbitrary user-specified feedback c.

The authors also introduce an online bootstrap procedure that conditions learning on positive feedback. The model parameters are updated through a two-step process: first, sampling an output given positive feedback, and then sampling new feedback based on that output.

Experimental evaluations are conducted on  mathematical and general reasoning tasks, while feedback signals are produced by GPT-5-nano.  Two baselines are considered: Rejection Sampling Finetuning (RFT) and GRPO.

**Strengths:**

- Training LLM with non-scalar reward is a very challenging task and of particular interest to avoid losing information and correct ambiguity of LLMs outputs in human-machine interactions.

- The proposed method is well motivated and the online procedure allows to consistently improve the offline trained policy.

- The experiments show that the proposed method offer performance that align with GRPO (and may slightly outperform GRPO in some cases).

**Weaknesses:**

- The experiments display mean accuracy for the various datasets making it hard to asses the statistical significance of the proposed method. The variability of the results over independent runs should be given.

-  The paper sets the focus on the fact that the core of RL is the online interaction with the environment and provides an algorithm which may reach slightly better performance than scalar-based RL approaches. However, it is not clear if the proposed feedback-based online method can provide consistent performance (if I understand well Table 1 & 2 for instance). When the proposed approach provides a lower accuracy, can it be improved by processing the feedback differently, optimizing the FCP differently etc.

I do not think the objective of such a paper is to  achieve performance that surpasses necessarily the current state of the art, but rather to gain a deeper understanding of the benefits of the proposed methodolgy of the policy optimization and to disentangle its genuine methodological contributions from implementation effects.

The method is interesting but I am not sure what is the main message except that the online boostraping seems to consistently improve the offline method.

**Questions:**

- The choice of the reverse KL is a natural choice and allows a straightforward optimization. However, other choices could be used for instance to balance the dissymetry of the KL divergence. Did you investigate optimizing other divergences ? Do you have empirical insight on the impact the way the FCP is trained has on the overall performance ?

- Tables provide accuracy to compare the different methods. I believe it would be important to add std over independent runs of the experiments to assess the statisstical significance of the proposed performance between methods.

- The experiment  using real-world user and professional reviewer style shows that the approach with user-style feedback offer an interesting gain with the base model but the professional style improves only slightly the performance. Does this suggest that there is room for improvement in the way feedbacks are preprocessed to improve the performance of the model ?

- In classical LLMs usage, human interaction is iterative to improve the performance of the output. Is it possible to extend the proposed approach by incorportating several iterative rewards in the online training to mimic the classical behavior of human-LLMs interaction ?

---

> ### Author Response · Authors · 2025-11-21
> **Rebuttal by Authors [1/2]**
>
> Thank you for your supportive review and suggestions. Below we respond to the comments in **Weaknesses (W)** and **Questions (Q)**.
>
> ---
>
> ***W1 & Q2: The variability of the results over independent runs should be given (such as standard deviation) in the Table.***
>
> Following your suggestion, we have added standard deviations to all results in $\\textrm{\\color{blue}Table 1}$ and $\\textrm{\\color{blue}Table 2}$ in the paper revision. As shown in the revised tables, the standard deviations for most mean accuracy values are below 1. The evaluation results indicate that the variance across independent runs is small and that the overall trends are stable. These statistics support our conclusion that FCP+Bootstrap achieves performance comparable to existing scalar-RL baselines.
>
> ---
>
> ***W2 & Q1-2 & Q3: Any empirical insights on the impact the way FCP is trained has on the overall performance? Such as processing the feedback differently, optimizing the FCP differently etc.?***
>
>
> Thank you for raising this question. Your intuition is correct: the performance of FCP can indeed improve when the feedback is processed differently, and in particular, the **feedback quality** is a key factor. To study this, we conduct controlled experiments using feedback with different semantic richness levels, from very low to high quality. The four settings we evaluate are:
>
> - **Correctness-Only (very low quality).** We use *math-verify* to check whether the model output matches the ground truth. If correct, the feedback is the string “correct”; otherwise “incorrect”. This provides almost no semantic information beyond a binary signal.
>
> - **Real-World User (low quality).** We use colloquial and subjective user-style feedback, such as “That looks right to me, concise and easy to follow. I’m satisfied with the final result.”
>
> - **Professional Reviewer-Lite (medium quality).** We prompt GPT-5-nano to give concise critique using only adjectives and adverbs, for example “correct, clear, concise, straightforward.”
>
> - **Professional Reviewer (high quality).** High-quality free-form critique such as “Correct and clear; succinct and logically sound, with concise and effective reasoning.”
>
> The results in $\\textrm{\\color{blue}Table 9}$ of paper revision **show a clear positive correlation between feedback quality and model performance.** For example, the average accuracy is **31.6** with correctness-only feedback, **32.8** with real-world user feedback, **34.2** with reviewer-lite feedback, and reaches **38.7** with full professional reviewer feedback. This progression reflects the increasing semantic richness of the critiques. High-quality verbal feedback conveys multiple evaluative dimensions, such as clarity, logical rigor, and completeness, which provide substantially stronger learning signals than correctness-only labels.
>
> Regarding the possibility of improving performance by optimizing FCP differently, **we also examine several design choices that may impact performance.** As discussed in Section 4.2 and $\\textrm{\\color{blue}Table 5}$ of paper revision, we ablate the online update strategy, the loss aggregation method, the batch size, etc. Our findings show that the semi-online update scheme yields the largest gains, which is why we adopt it as our main training setup.  We believe that additional improvement is possible, and we plan to further explore optimization strategies for FCP in future work.
>
> ---
>
>
> ***Q1-1: Investigation to optimizing other divergences.***
>
> We thank the reviewer for raising this point. In principle, many other divergences could also be considered. We chose the KL divergence mainly because, under our formulation, it leads to a simple likelihood-based objective that can be optimized directly with standard supervised learning procedures. Using other divergences would typically require estimating density ratios or introducing additional auxiliary models, which is considerably more complex and often less stable in the conditional generation setting we study.
> We did not investigate alternative divergences in this work. Conceptually, however, our approach does not depend on a specific divergence: the KL is selected because it provides a favourable combination of tractability and stable optimization, and already yields strong empirical results. We will continue to explore alternative divergences as part of our future research.

---

> ### Author Response · Authors · 2025-11-21
> **Rebuttal by Authors [2/2]**
>
> ---
>
> ***W3: What is the main message except that the online bootstrapping seems to consistently improve the offline method.***
>
> We thank the reviewer for the thoughtful comments. Our main message is that **FCP serves as a general learning paradigm for the era of environment feedback**.
>
> As discussed in recent work on the era of experience [1], the supply of high-quality human-generated data for pretraining is predictably approaching saturation, and even verifiable data, while reliable, is fundamentally limited in scale and diversity. As a result, the primary source of new training signals for capable agents will increasingly come from *interaction with environments* and the *feedback* that such interactions produce.
>
> Rather than proposing another variant of reward-based RL, our goal is to frame FCP as a learning paradigm that *directly models the structure of environment feedback without compressing it into scalar rewards*. **This perspective is not specific to language models**. The framework applies whenever an agent receives observable feedback $c$ from an environment in response to an action or output $o$. In this sense, FCP complements scalar-reward RL as a parallel paradigm that is naturally suited for learning from rich, structured environment feedback.
>
> In addition, there are several theoretical advantages that are feasible in principle and open promising future directions:
> - **Compatibility with verifiable scalar rewards.** FCP can incorporate verifiable rewards by using a **null feedback condition** (e.g., <EF></EF>) when performing scalar RL optimization.
> - **Applicability to any modality.** FCP builds on $\\pi\_\\theta(o|x,c) \\propto \\pi\_{\\mathrm{ref}}(o|x)\\cdot p\_{\\mathrm{env}}(c|x,o)$, and places no constraints on the modalities of  $x$, $o$, $c$, which can be interleaved text/images/videos/audios.
> - **Test-time adaptation to individual feedback styles.** FCP enables rapid adaptation to user-specific feedback styles at inference time by conditioning on a small number of user-provided examples.
>
> [1] Welcome to the Era of Experience, 2025
>
> ---
>
>
>
> ***Q4: Is it possible to extend the proposed approach by incorporating several iterative rewards in the online training to mimic the classical behavior of human-LLMs interaction?***
>
>
> We are very glad that you raised this point because your suggestion aligns closely with the direction we plan to explore next. In real multi-turn human-LLM interactions, users naturally provide iterative verbal feedback, such as “the first paragraph is unclear, please rewrite it.” In this example, *“the first paragraph is unclear”* serves as explicit critique, while *“please rewrite it”* is a revision instruction. Leveraging such naturally occurring conversational signals would allow the model to learn in a more human-centric and aligned manner.
>
> Our current work focuses on establishing that, even in the single-turn setting, **FCP achieves performance comparable to mainstream scalar-RL methods** on reasoning tasks without relying on scalar rewards. Building on this foundation, we see the multi-turn setting as a natural and promising extension.
>
> A straightforward way to incorporate iterative feedback into FCP is as follows. Consider a multi-turn dialogue trajectory $\\{x\_1, o\_1, x\_2, o\_2, \\ldots, x\_T, o\_T\\}$, where $x\_i$ is the user’s input at turn $i$, and $o\_i$ is the model’s response. For each turn $i$, we obtain verbal feedback $c\_i$ using one of two mechanisms:
>
> - **Extracting feedback from the next user message.** If $x\_{i+1}$ contains feedback that can be cleanly separated from instructions (for example, *“the first paragraph is unclear”*), we treat that segment as verbal feedback $c\_i$.
>
> - **Simulating feedback when the next message contains no critique.** If $x\_{i+1}$ does not include any usable feedback, we query an LLM judge to produce $c\_i \\sim p\_{\\text{env}}(c | x\_i, o\_i)$.
>
> The training objective becomes $\\max\_{\\theta} \\mathbb{E}\\left[ \\log \\pi\_{\\theta}(o\_i | x\_1, o\_1, \\ldots, x\_i, c\_i) \\right]$, so the model is conditioned on both the interaction history and the verbal feedback for each turn.
>
> At test time, the process remains simple and efficient. The model does not require iterative feedback from any environment. We can automatically obtain a suitable positive feedback description by prompting an LLM (for example, “Provide several possible positive feedback descriptions for the following query.”). The model then conditions on this feedback description and generates the final output in a single pass, without any multi-step refinement during inference.
>
> Overall, extending FCP to learn from iterative verbal feedback in multi-turn interactions is a natural and promising direction. Your question aligns directly with our ongoing plans, and we appreciate the opportunity to clarify this potential extension.

---

> > ### Comment · Reviewer_8GhM · 2025-11-26
> >
> > I would like to thank the authors for their detailed answers and comments.
> > I appreciate the additional experiments on the feedback processing and the controlled experiments using feedback with different semantic richness levels.
> > I also believe that your comments on Q4 could be discussed in the article to highlight better the potential perspectives.

---

> > > ### Author Response · Authors · 2025-11-26
> > > **Thank you for your support and raising the score**
> > >
> > > Thank you for your thoughtful feedback and for engaging deeply with our work, we greatly appreciate it.
> > >
> > > In the final revision, we will incorporate your suggestions by (1) reporting standard deviations for all results, (2) further analyzing the impact of feedback processing and feedback quality on performance, and (3) improving the writing to state the main message more clearly. We will also expand the discussion of multi-turn iterative feedback (Q4) in the article, as you suggested, to more explicitly highlight this future direction.
> > >
> > > Thank you again for your constructive review. It was a pleasure to discuss these points with you, thank you!

---

### Official Review · Reviewer_4wXq · 2025-10-29

**Soundness:** 3
**Presentation:** 1
**Contribution:** 2
**Rating:** 2
**Confidence:** 4

**Summary:**

This paper proposes a new feedback-based training strategy for Large Language Models (LLMs), based on using verbal feedback rather than the conventional real value, reward based, feedback.

Their theoretical justification follows previous approaches using optimal policy behavior cloning. These approaches use following facts

- If we consider the objective of minimizing the expected reward (in this case log probability of feedback c) constrained to be close to some reference policy via KL divergence, it has a closed form solution in the form of an energy model (optimal policy)

- that optimal policy has an intractable partition function, since it requires summing over the countably infinite space of sentences. Sampling from that policy via rejection sampling or importance sampling is however simple

- rather than directly solving that objective with an amortized policy and gradient descent as PPO/GRPO do, or deriving a related optimization problem as DPO does, some other approaches (such as DPG, GDC, GDC++ and BRAIn) just sample from that optimal policy and do behavior cloning

What aspects are different in this new work

- use of verbal feedback, previous approaches use conventional rewards, which blends well with optimal policy behavior cloning
- the do not amortize, previous approaches just train a model to predict the optimal policy output given the input, but do not condition on the reward/feedback
- this brings a fundamental complication, now we need feedback at test time, which is defined as conditioned on an input and an output. There are additional changes to account for this including using a model that provides feedback and a second tuning stage

**Strengths:**

1. The idea of using verbal feedback is not exactly new e.g. [1] but feels like a very natural evolution of current trends. The authors also make a compelling case for it

2. The authors also manage to connect it with some well established optimization principles related to optimal policy.

3. Results seem strong against well know baselines like GRPO, although I have my concerns about the setup

[1] RLVF: Learning from Verbal Feedback without Overgeneralization, 2024

**Weaknesses:**

1. Unfortunately presentation is not at the same level of the rest of the paper. The authors dedicate a lot of space to make a case for the use of non-verbalized rewards, including quoting Sutton in a separate paragraph. Then declare "Due to space constraints, related work is deferred to Appendix B.". This is beyond some formatting issue. Related work is central to a paper and can not moved to an optional section such as appendix. It is great if the authors want to include an extended related work section in the appendix, but the core relevant works should be in the body.

2. The paper could also be more clear. Maybe is the way I understand the method, but the question of how do you operate at test time (since you will need feedback) is central and is only addressed implicitly and later in the paper. On has to read the algo descriptions to understand how this problem is solved. I could not follow as well the justification of why the negative feedback can be used for learning. I think the analogy to composition of concepts in image to text may make the idea harder to follow.

3. The paper also misses a line of work that directly overlaps with this work. The claim "we instead propose to minimize the forward KL divergence [...]" is not correct. This has already been proposed in other works solving the KL-constrained reward maximization problem described in the summary (which is reverse KL) with a forward KL approximation (see my summary for the differences). I would say at least [1] who aplies DPG to LLMs for style control, [2] who brings the same point about the optimal policy made here and brings it closer to RL of LLMs and [3] which directly formulates it as Bayesian optimization, are relevant here

[1] A Distributional Approach to Controlled Text Generation, 2021
[2] On reinforcement learning and distribution match- ing for fine-tuning language models with no catastrophic forgetting, 2022
[3] BRAIN: Bayesian Reward-conditioned Amortized INference for natural language generation from feedback, 2024

I do think the authors can make a convincing case for fixing all of the above (see questions).

4. however, there are also concerns with the experimental setup. If I understood this correctly, human feedback is simulated using gpt-5-nano, which is particularly strong in math and instruction following (used in the setup). While all methods have access to this judge as a provider of either scalar reward or verbal feedback, only the proposed method here can use this feedback as input to the prompt.

That is you are using gpt-5-nano to provide some hint in the input to your model, which is trained to utilize this hint from this specific model. This is a big advantage and would require the use of an additional, powerful model at test time. If this is the case then I do not think the comparison in fair.

Further, forward-KL optimal policy behavior cloning approaches are not used nowadays, which implies that they are not very competitive. This is just a hunch but this may further point to the fact that the hint from a stronger model is behind the improvements.

**Questions:**

How would you address weaknesses 1-3?

Is my understanding of the use of gpt-5 in weakness 4.  correct?

---

> ### Author Response · Authors · 2025-11-21
> **Rebuttal by Authors [1/2]**
>
> Thank you for your valuable review and suggestions. Below we respond to the comments in **Weaknesses (W)**.
>
> ---
>
> ***W1: The core related works should be given in the body.***
>
> Thank you for pointing out this issue. In the paper revision, we have moved the Related Work section into the main body (*the page limit increases to 10 during rebuttal*). We have also revised the section to ensure clearer coverage, comparison, and positioning of our contributions relative to prior work.
>
> ---
>
> ***W2-1: Unclear test-time operation.***
>
> Following your suggestion, we clarify this point in the paper revision. **In Section 2.2, we add a dedicated paragraph titled “Test-time usage”**, which explicitly explains how the model operates without requiring environment-generated feedback during inference. The revised text states that the user may optionally provide a desired feedback condition, and that when no such condition is given, a positive feedback description is automatically obtained by prompting an auxiliary LLM (e.g., “Provide several possible positive feedback descriptions for the following query.”). The model then generates the final output in a single forward pass process.
>
> ---
>
> ***W2-2: The justification of why the negative feedback can be used for learning.***
>
> In our framework, negative feedback is not treated as a reward but simply as an instance of the condition variable $c$ in the conditional policy $\\pi\_\\theta(o|x, c)$. For a negative example $(x, o\^{-}, c\^{-})$, the optimal feedback conditional policy has the form:
> $$
> \\pi\_\\theta(o\^{-} | x, c\^{-}) \\propto \\pi\_{\\mathrm{ref}}(o\^{-} | x)\\cdot  p\_{\\mathrm{env}}(c\^{-}| x, o\^{-})
> $$
> which directly encourages the model to learn **what types of responses are associated with negative critiques**. Crucially, because at test time the policy under a positive conditioning $c\^{+}$ is $\\pi\_\\theta(o \\mid x, c\^{+}) \\propto \\pi\_{\\mathrm{ref}}(o\\mid x)\\, p\_{\\mathrm{env}}(c\^{+}\\mid x,o)$, then any output $o\^{-}$ for which $p\_{\\textrm{env}}(c\^{+}\\mid x, o\^{-}) \\ll p\_{\\textrm{env}}(c\^{-}\\mid x, o\^{-})$ will **naturally receive very small probability mass** when conditioned on $c\^{+}$. Thus, negative examples provide information about the region of the output space that should be *avoided* when conditioned on positive feedback, because:
> $$
> \\pi\_\\theta(o\^{-} \\mid x, c\^{+}) \\propto p\_{\\mathrm{env}}(c\^{+}\\mid x, o\^{-}) \\approx 0
> $$
>
> In other words, learning from $(x,o\^{-},c\^{-})$ teaches the model **which outputs are incompatible with the semantics of positive feedback $c\^{+}$**, thereby eliminating those regions of the output space during inference. This effect follows directly from the conditional factorization used throughout the paper, with no additional assumptions.
>
> A second key point is that our method **doesn't require distinguishing whether a feedback $c$ is positive or negative**. The model simply learns the joint structure of $(o, c)$ through the conditional distribution $\\pi\_\\theta(o |x, c)$: $c$ is treated uniformly as a conditioning signal, and the training objective does not depend on any binary label of the feedback. Therefore, negative feedback is “usable” not because we explicitly interpret its sentiment, but because the model learns how *different types of feedback correspond to different regions of the output space*.
>
> ---
>
> ***W3: The claim "we instead propose to minimize the forward KL divergence [...]" is not correct. This has already been proposed in other works.***
>
> We would like to clarify that we never intend to position the use of forward KL as a contribution of our work. Forward KL is such a standard and widely used probabilistic metric, and in our paper it serves purely as a methodological choice rather than a novel component. Our original phrasing (“we instead propose to minimize the forward KL…”) may have unintentionally suggested otherwise.
>
> To avoid any misunderstanding, we have revised the corresponding sentence in the main text to: `“… we instead turn to minimize the forward KL divergence. This relaxation is standard in KL-constrained LM training, where forward-KL objectives have been used for distributional or reward-conditioned generation [1,2,3], though our setting differs in conditioning on rich verbal feedback.”`
> This revision acknowledges the existing literature while clarifying that our contribution lies in learning directly from verbal feedback rather than in the choice of KL direction.
>
> [1] A Distributional Approach to Controlled Text Generation, 2021
>
> [2] On reinforcement learning and distribution matching for fine-tuning language models with no catastrophic forgetting, 2022
>
> [3] BRAIN: Bayesian Reward-conditioned Amortized INference for natural language generation from feedback, 2024

---

> > ### Comment · Reviewer_4wXq · 2025-11-23
> >
> > >Forward KL is such a standard and widely used probabilistic metric
> >
> > Than you for your clarification and edits. To be completely clear, my criticism is not about the "use of forward-KL" in general, but rather the use of forward-KL as a an alternative to solving the equivalent reverse-KL problem in KL-controlled reward maximization. Your update solves this issue now.

---

> > > ### Author Response · Authors · 2025-11-24
> > > **Thank you for your support and raising the score**
> > >
> > > Thank you for your timely feedback and kind words. We greatly appreciate it!
> > >
> > >
> > > In the final revision, we will incorporate your suggestions by further clarifying why negative feedback can be used for learning. We will also refine the discussion of the forward-KL formulation to better distinguish it from the reverse-KL perspective. In addition, we will extend the weaker-model analysis to more directly evaluate how critique model capability affects the resulting feedback-conditional policy.
> > >
> > >
> > >
> > > Thank you again for your insightful review, which has greatly inspired us to improve our work. It was a pleasure to discuss these points with you, thank you!

---

> ### Author Response · Authors · 2025-11-21
> **Rebuttal by Authors [2/2]**
>
> ---
>
> ***W4: Usage of verbal critiques from GPT-5-nano may bring unfair comparison.***
>
> To address this concern directly, we run an additional experiment where the critique model is replaced with a **much weaker** model, `Qwen2.5-1.5B-Instruct`. This experiment directly tests whether the performance gains are attributable to the critique model’s strength or to the FCP framework itself.
>
> As shown in $\\textrm{\\color{blue}Table 10}$, using `Qwen2.5-1.5B-Instruct` as the provider of user-style feedback yields an average accuracy of **33.1**, which is **comparable to the **32.8** obtained when using GPT-5-nano** as the critique model, and significantly higher than the base model’s 27.0. This demonstrates that **FCP does not rely on a strong critique model in order to achieve meaningful improvements**.
>
> Furthermore, `Qwen2.5-1.5B-Instruct` itself performs far worse on the benchmarks (average accuracy **19.2**) than the model being trained, `Qwen2.5-7B-Base`(**27.0**). This clearly rules out the possibility that FCP is simply “learning hints” from a stronger model. The critique model in this setting is strictly weaker, yet FCP still improves substantially over the base model.
>
> Additionally, $\\textrm{\\color{blue}Table 1}$ in the paper includes a CFT baseline[1], where the model is trained directly on the same $(x, o, c)$ triples by supervised fine-tuning on $p(c | x, o)$, that is, directly mimicking the critiques produced by GPT-5-nano. This setting performs very poorly, reaching an average accuracy of only **9.1**, far below the base model’s **27.0**. This result further indicates that **the critiques themselves do not contain learnable knowledge that can be exploited through direct imitation**.
>
> [1] Critique Fine-Tuning: Learning to Critique is More Effective than Learning to Imitate

---

> > ### Comment · Reviewer_4wXq · 2025-11-23
> >
> > This result is helpful thanks. I will improve my score accordingly.

---

### Official Review · Reviewer_hpxd · 2025-11-01

**Soundness:** 3
**Presentation:** 3
**Contribution:** 3
**Rating:** 6
**Confidence:** 3

**Summary:**

This paper proposes a new framework, Feedback-Conditional Policy (FCP), for training large language models (LLMs) directly from verbal feedback rather than scalar rewards. Instead of reducing nuanced feedback into numerical values, the authors treat feedback as a conditioning signal and learn the posterior π_θ (o∣x,c)

The approach includes an offline training (learning from response, feedback pairs) and online bootstrapping (conditioning on positive feedback to improve iteratively). Experiments on mathematical and general reasoning benchmarks show that FCP achieves competitive or better performance than scalar-based baselines such as RFT and GRPO, without relying on verifiers or reward models.

**Strengths:**

1. Learning from verbal feedback is an important direction for scaling alignment. The authors clearly articulate why scalarization leads to information loss and imbalance. With LLMs being deployed at such large scale, collecting descriptive feedback can reveal the precise nature of model issues, rather than relying on simplistic scalar rewards (such as thumbs up or thumbs down).


2.  The distinction between real-world users and professional reviewers is particularly valuable, and the corresponding ablation study effectively illustrates the interplay between these two feedback sources.

3.  The experimental setup is well designed and clearly demonstrates the effectiveness of the proposed approach, particularly through the analysis of different feedback conditions.

**Weaknesses:**

1. Generally, it is costly to collect verbal feedback from humans for training, and hence FCP can be used only with LLM as the environment that is capable of giving both scalar and verbal feedback.

2. In Algorithm 2, $p_{user}(c+)$ is one of the inputs that is used to sample $c+$. It is done on the basis of scalar rewards (scalar scores) obtained in the offline phase (line 300 to 303). That is, the method assumes access to an environment that provides both verbal and scalar feedback (to construct $p_{user}$). However, Table 1 claims that FCP+bootstrap doesn’t require a scalar reward – this seems inaccurate. You use the accompanying scalar reward for sampling {c+} from $p_{user}$.

3. Typically, models are finetuned by experts and used by others in various scenarios. Often, these users are oblivion to the training details. I wonder how do you except users to provide their inputs along with the “desired feedback”. Further, the user feedback could be from a different distribution than the one used for training, whereas in all experiments, you sample a {c+} from the ones obtained during training. Maybe it's worthwhile to report numbers in a scenario where c+ is sampled from a different distribution.

4. The assumption that users will provide explicit, well-formed positive feedback (e.g., “accurate and concise reasoning”) during inference may not be realistic. I believe it is crucial to investigate how FCP behaves when presented with no feedback or ambiguous feedback as input.

5. While it is reasonable to use scalar rewards from GPT-5-nano for experimental comparison, to more convincingly demonstrate the effectiveness of verbal feedback, it would be valuable to include experiments using *verifiable feedback* (e.g., correctness-based signals). The goal need not be to outperform such setups, but rather to quantify how close verbal feedback learning comes to verifiable feedback performance.

**Questions:**

When testing on the trained task, the input prompt is not just the query x, but also feedback (wrapped in special tokens <EF>). How is the prompt created for OOD tasks?

---

> ### Author Response · Authors · 2025-11-21
> **Rebuttal by Authors [1/2]**
>
> Thank you for your supportive review and suggestions. Below we respond to the comments in **Weaknesses (W)** and **Questions (Q)**.
>
> ---
>
> ***W1: FCP can be used only with LLM as the environment, as human feedback is costly to collect.***
>
> Collecting verbal feedback from humans could be costly *for a research project*. Given this constraint, our current experimental setup uses LLMs as the alternative to collect verbal feedback.
>
> However, in real-world **production environments**, there is already **a large amount of human verbal feedback** collected either online or offline. Systems such as ChatGPT, Gemini, and other large-scale chatbot platforms continuously collect real user feedback during daily interactions. With light filtering and cleaning, this data can be used directly for our FCP training. In such settings, FCP would learn from genuine human preferences and become even more aligned with how users naturally evaluate model outputs. We believe this represents a highly practical and impactful deployment direction.
>
> Furthermore, FCP is **not more expensive than existing approaches**. FCP is naturally more suitable for domains without verifiable ground truth, such as open-ended dialogue and creative writing. In these areas, current practice already relies on reward models, typically LLMs, to serve as the evaluation environment. These reward models produce both chain-of-thought reasoning and scalar scores, and the reasoning itself contains rich verbal feedback. As a result, the cost profile of FCP is essentially identical to that of existing RLHF or RLAIF pipelines that also rely on LLM-based evaluators.
>
> ---
>
>
> ***W2: The usage of scalar score in offline phase to help sample $c\^+$ is controversial to the claim that FCP+bootstrap doesn’t require a scalar reward.***
>
> Regarding the use of scalar scores in the offline phase, this is only a **convenient data filtering mechanism and is entirely optional**. In real applications, $c\^+$ can be obtained without any scalar information. For example, one can simply prompt an LLM with instructions such as *“Provide several possible positive feedback descriptions for the following query.”* to directly generate positive critiques. This avoids scalar scores altogether and produces $c\^+$ through straightforward LLM prompting.
>
>
> ---
>
>
> ***W3-1: How do you expect users to provide their inputs along with the “desired feedback”?***
>
> Yes, end-users are typically unaware of the model’s training details and should not be expected to provide their own “desired feedback” at test time. Instead, in practical deployment scenarios, an appropriate positive feedback $c\^+$ could be automatically generated from *a lightweight auxiliary workflow* based on the user’s prompt. For example, for writing tasks, it might infer expectations like “well-structured, factual” rather than simply “correct”. This mechanism is inspired from existing text-to-image systems, where user prompts will be augmented by LLMs before being fed into text-to-image models.
>
> ---
>
>
> ***W3-2 & W4: How does model behave if $c\^+$ is sampled from a different distribution? And how does FCP behave when presented with no/ambiguous feedback?***
>
> Following your suggestions, we also evaluate the model under several feedback settings that are out-of-distribution relative to the training data. Specifically, we consider:
> - a *no-feedback* setting, where the expected feedback is an empty string;
> - a simple *“Correct”* setting, where the expected feedback consists of a single word;
> - a *Real-World User* setting, where we use colloquial, first-person user-style feedback. For example, “That looks right to me, concise and easy to follow. I’m satisfied with the final result.”
>
> The results in $\\textrm{\\color{blue}Table 8}$ of our paper revision give two clear observations. First, across all OOD settings, **the model still shows large gains over the base model**. For example, even with *no feedback*, FCP+Bootstrap reaches an average score of **34.7**, compared with **27.0** for the base model. Similarly, with *Real-World User* feedback, FCP+Bootstrap obtains **33.3**, and with “Correct” feedback it reaches **34.4**. These results show that **the method remains strong even when the feedback distribution shifts substantially**.
> We hypothesize that this robustness arises because the system and user prompts already contain implicit expectations of positive behavior (for instance, the default system prompt “You are a helpful assistant”). As a result, the model continues to benefit even when no explicit or in-distribution feedback is provided.

---

> ### Author Response · Authors · 2025-11-21
> **Rebuttal by Authors [2/2]**
>
> ---
>
> ***W5: It would be valuable to include experiments using verifiable feedback (e.g., correctness-based signals).***
>
> Following your suggestion, we add three additional baselines that use verifiable feedback. We include the following settings:
>
> - **GRPO with rule-based verifier.** We use the math-verify package as a symbolic correctness checker. The model receives a scalar reward of 1 when its answer matches the ground truth and 0 otherwise;
>
>
> - **FCP+Bootstrap with rule-based verifier.** Using the same math-verify checker, the model receives the verbal feedback “correct” or “incorrect” instead of a scalar reward.
>
>
> As shown in $\\textrm{\\color{blue}Table 7}$ of our paper revision, GRPO with rule-based verifier attains an average score of **36.4**, which is not higher than GRPO using LLM-generated scalar rewards (**38.4**). A similar pattern appears for RFT+GRPO, where rule-based signals give **37.8**, compared with **37.1** for LLM-based signals. This suggests that a symbolic correctness signal does not offer a clear advantage over LLM-based scalar rewards in our setting. We believe two factors contribute to this outcome:
>
> - First, the **LLM-generated scalar scores compress multiple critique dimensions into a single value**, and dimensions such as logical rigor, completeness, and clarity provide additional learning signals beyond correctness alone.
>
> - Second, rule-based verifiers frequently encounter **prompts where all sampled responses receive identical scores, which prevents effective GRPO updates** [1]. This situation rarely occurs with LLM provided scores, which can take eleven distinct values ranging from 0 to 10.
>
> Regarding verbal feedback, FCP+Bootstrap trained with only “correct” or “incorrect” labels reaches an average score of 31.6, which is higher than the base model (27.0) but notably lower than the setting using richer LLM-generated verbal feedback (38.7). This highlights the importance of multi-aspect critiques rather than correctness alone.
>
> Taken together, these results show that FCP+Bootstrap trained with verbal feedback achieves performance comparable to, and in some cases slightly better than, setups that rely on verifiable correctness-based signals, further supporting our claim that **verbal feedback learning is a competitive and effective approach**.
>
> [1] DAPO: An Open-Source LLM Reinforcement Learning System at Scale
>
>
> ---
>
> ***Q1: How is the prompt created for OOD tasks?***
>
> For OOD tasks, we obtain a positive feedback $c\^+$ at inference time by prompting an LLM, e.g., with instructions such as “Provide several possible positive feedback descriptions for the following query.” and then selecting one of the generated candidates. Users may also supply their own preferred feedback condition if desired.
> In practice, the prompt is constructed by placing the chosen feedback description before the query, for example:
> `<EF>Correct and clear, with well-structured reasoning.</EF>original_prompt`
> We clarify this test-time usage explicitly in the updated PDF.

---

### Official Review · Reviewer_pzbE · 2025-11-01

**Soundness:** 3
**Presentation:** 3
**Contribution:** 3
**Rating:** 6
**Confidence:** 3

**Summary:**

This paper proposes to learn from the verbal feedback directly to avoid collapsing rich verbal feedback into a single scalar reward. It treats feedback as a conditioning signal and trains a feedback-conditional policy (FCP) that models $\pi_\theta(o \mid x, c) \propto \pi_{\mathrm{ref}}(o \mid x)\, p_{\mathrm{env}}(c \mid x, o)$. Concretely, they (1) collect an offline dataset of (instruction $x$, model response $o$, feedback $c$) triples from a reference policy plus a feedback generator, and maximize log-likelihood of the response given the feedback, and then (2) do an online bootstrapping stage where they condition the current model on positive feedback, roll out, get fresh verbal feedback, and train again on those newly labeled triples. This reframes “learning from feedback” as conditional generation rather than reward optimization. Experiments on math and general reasoning tasks show that offline FCP is  comparable to RFT, and FCP + bootstrapping reaches or slightly surpasses strong scalar baselines like GRPO/RFT+GRPO.

**Strengths:**

Learning directly from rich, multi-aspect LLM feedback instead of collapsing it into a single scalar reward is an interesting and important problem that is relatively new to the LLM era. This paper offers a conceptually novel solution to this problem, although the empirical gains over scalar-based baselines are not yet significant.

**Weaknesses:**

1. The empirical performance gains over baselines are not significant.
2. The paper mostly compares against basic scalar/fine-tuning baselines (e.g., GRPO/CFT), but does not evaluate against recent critique-based methods such as Critique-GRPO: Advancing LLM Reasoning with Natural Language and Numerical Feedback.

**Questions:**

In Sec. 2.1 you make two approximations: (1) you approximate $p_{\text{off}}(o \mid x, c^+)$, and (2) you replace the reverse KL in the ideal objective with a forward KL. Could you clarify the impact of these approximations?

---

> ### Author Response · Authors · 2025-11-21
> **Rebuttal by Authors**
>
> Thank you for your supportive review and suggestions. Below we respond to the comments in **Weaknesses (W)** and **Questions (Q)**.
>
> ---
>
> ***W1: The empirical performance gains over baselines are not significant.***
>
> Following your suggestion, we conducted additional experiments, and the resulting empirical gains become even clearer.
>
> As shown in $\\textrm{\\color{blue}Table 7}$, we now include two strong baselines that rely on **verifiable, rule-based correctness signals**. These baselines represent settings where ground-truth checking is available, which is generally considered advantageous because correctness verification offers clean and deterministic feedback. The rule-based GRPO variants achieve average scores of **36.4** for Base+GRPO and **37.8** for RFT+GRPO. Our method, FCP+Bootstrap trained purely with LLM-generated verbal feedback, reaches **38.7**, which is **highly competitive with these correctness-based RL pipelines** and slightly exceeds their average performance.
>
> Beyond the empirical results, our approach also provides **several advantages that scalar pipelines lack**. These include avoiding the need to filter out responses with zero reward variance, the ability to learn from partially correct samples, and finer-grained controllability at inference time by conditioning on expected feedback.
>
> ---
>
> ***W2: Lack of comparison to recent critique-based methods such as Critique-GRPO.***
>
> Following your suggestion, we have started running the corresponding experiments comparing recent critique-based methods such as Critique-GRPO [1]. Due to time constraints, these runs are still in progress, and we expect to update the results within the coming weeks before rebuttal ends.
>
> [1] Critique-GRPO: Advancing LLM Reasoning with Natural Language and Numerical Feedback
>
> ---
>
> ***Q1-1: Clarification for the impact of $p\_{\\text{off}}(o|x,c\^+)$ approximation (in Sec. 2.1).***
>
> In Sec. 2.1, our “ideal” target is the feedback-conditional posterior $P\_{\\text{off}}(o |x, c\^+)$ in Eq. (1–2), which assumes access to the exact environment likelihood $p\_{\\text{env}}(c\^+ |x, o)$ and to a training dataset containing only purely positive feedback $c\^+$.
>
> In practice, however, we can only samples $c \\sim p\_{\\text{env}}(\\cdot |x, o)$, and we approximate this posterior with a parametric policy $\\pi\_\\theta(o |x, c)$. The approximation is therefore: instead of directly optimizing Eq. (2) for the single condition $c\^+$, we learn $\\pi\_\\theta(o |x, c)$ to match $P\_{\\text{off}}(o |x, c) $ for all observed $c$, and at test time we simply plug in a user-defined positive feedback $c\^+$.
>
> Crucially, this **does not** change the target distribution for a fixed feedback condition: the global optimum of the objective in Eq. (3) is attained when $\\pi\_\\theta(o |x, c) = P\_{\\text{off}}(o |x, c)$ for every $c$ (including $c\^+$). Thus, in the infinite-data / infinite-capacity limit, conditioning on $c = c\^+$ recovers exactly the “ideal” posterior $P\_{\\text{off}}(o |x, c\^+)$.
>
> The practical impact of this approximation is therefore limited to standard finite-sample and function-class approximation error, which is common to all parametric methods. Empirically, Sec. 3 shows that the resulting FCP achieves performance comparable to or better than scalar-based baselines, suggesting that this approximation is not detrimental in practice.
>
> ---
>
> ***Q1-2: Clarification for the impact of replacing the reverse KL in the ideal objective with a forward KL (in Sec. 2.1).***
>
> Eq. (2) can be written as minimizing the *reverse KL divergence* $D\_{\\mathrm{KL}}(\\pi(o |x, c\^+) || P\_{\\text{off}}(o |x, c\^+))$, but this form is **intractable** because it requires computing $\\log p\_{\\text{env}}(c\^+ |x, o)$ for candidate responses $o$. To obtain a tractable training objective *with the same optimal solution*, we instead minimize the *forward KL divergence* $D\_{\\mathrm{KL}}(P\_{\\text{off}}(o |x, c) || \\pi\_\\theta(o |x, c))$. This choice has two consequences:
>
> - **Theoretical:** the forward-KL and reverse-KL formulations share the *same* global optimum, namely $\\pi(o |x, c) = P\_{\\text{off}}(o |x, c)$. Thus, at the level of target distribution, we do not change what we are trying to approximate.
>
> - **Practical / behavioral:** forward KL is known to be more “mass-covering” than reverse KL, which is more “mode-seeking”. In our setting, $P\_{\\text{off}}$ is induced by a strong reference model and environment feedback, and the forward-KL objective reduces to standard maximum-likelihood training on $(x, c, o)$ triples. This gives a very simple, stable supervised objective while remaining statistically consistent with the ideal posterior. Empirically, we do not observe this to be detrimental: conditioning on positive feedback $c\^+$ yields sharp, high-quality responses and strong accuracy (as shown in $\\textrm{\\color{blue}Table 1}$ and $\\textrm{\\color{blue}Table 2}$).

---

> ### Author Response · Authors · 2025-11-27
> **Update on W2: Comparison with Critique-GRPO — Results and Analysis**
>
> As suggested, we added Critique-GRPO to $\\textrm{\\color{blue}Table 1}$. Following [1], our procedure retains only one refined response per prompt. Under the base model initialization, Critique-GRPO reaches an average math accuracy of **35.4**, lower than both **GRPO (38.4)** and our **FCP+Bootstrap (38.7)**. A similar pattern appears under RFT checkpoint initialization as well.
>
> We suspect this is due to critique/feedback in our setting being extremely **high-level**. As shown in $\\textrm{\\color{blue}Table 3}$, feedback such as `Accurate and clear; concise and coherent reasoning; correct conclusion`, mainly reflects correctness without offering step-wise guidance. Unlike the fine-grained CoT critiques leveraged in [1], such coarse feedback provides **limited useful signal for refinement**, and incorporating it into resampling likely introduces **additional noise** rather than helping the model improve reasoning. In contrast, FCP+Bootstrap remains effective without requiring detailed critique/feedback quality.
>
> [1] Critique-GRPO: Advancing LLM Reasoning with Natural Language and Numerical Feedback, 2025

---

### Author Response · Authors · 2025-11-21
**Summary of Paper Revision**

We thank all reviewers for their constructive feedback, and we have responded to each reviewer individually. We have also uploaded a **Paper Revision** including additional results and illustrations:
- $\\textrm{\\color{blue}Sec. 2.2}$ (Page 5): We add a detailed explanation of the test-time usage of FCP.
- $\\textrm{\\color{blue}Table 1}$ (Page 6): We include standard deviations to report the variability across runs, and have additionally added the Critique-GRPO [1] baseline for completeness.
- $\\textrm{\\color{blue}Table 2}$ (Page 7): We include standard deviations to report the variability across runs.
- $\\textrm{\\color{blue}Sec. 5}$ (Page 9): We move the related work section from the appendix into the main body.
- $\\textrm{\\color{blue}Table 7}$ (Page 18): Comparison between FCP and GRPO with rule-based verifier.
- $\\textrm{\\color{blue}Table 8}$ (Page 18): Evaluation results under out-of-distribution feedback conditions.
- $\\textrm{\\color{blue}Table 9}$ (Page 18): Effects of training-used feedback quality on FCP performance.
- $\\textrm{\\color{blue}Table 10}$ (Page 19): Comparison of FCP performance using weak critique models.

[1] Critique-GRPO: Advancing LLM Reasoning with Natural Language and Numerical Feedback, 2025

---

### Author Response · Authors · 2025-12-01
**Summary of Rebuttal Discussion**

We thank all reviewers for the valuable feedback. During rebuttal, we conducted additional experiments and made writing clarifications to address concerns raised by each reviewer.

Due to the unexpected early closure of the discussion phase, **Reviewers pzbE** and **hpxd** did not provide further follow-up comments after our responses.

**Reviewer 4wXq** confirmed that our rebuttal successfully addressed the initial concerns and stated ***“I will improve my score accordingly.”***

**Reviewer 8GhM** appreciated the additional experiments, particularly those on feedback processing and feedback with different semantic richness levels, and **subsequently raised the score.**

Overall, we believe the discussion progressed positively and that most concerns have been addressed with new results and clarifications.

---

### Meta-Review · Area_Chair_Y7fc · 2026-01-02

**Summary:**

The paper proposes a method to train LLMs from verbal feedback without explicit scalar rewards. The key difference from previous work, as shown in Eqn (10), is how conditioning is applied. In this sense, the novelty of this paper is limited, as neither formulation (forward dynamic or the proposed inverse dynamic) is wrong, nor theoretically proven to be superior than the other.

Since the paper is not quite related to RL, the RL-based narrative and the epigraph quoting the Reward Hypothesis are somewhat contrived.

Reviewers raised concerns mainly focusing on experimentation and the writing of the paper. In the author response, the authors provided some clarifications and shuffled some sections around, which appear to have addressed some of the concerns. For experimentation, the authors provided 6 additional tables, which is very impressive. But this also suggests that the paper should go through another round of thorough review to validate the massive additional experiments.

**Reviewer Concerns:**

Reviewers raised concerns mainly focusing on experimentation and the writing of the paper.

**Reviewer Scores:**

In the author response, the authors provided some clarifications and shuffled some sections around, which appear to have addressed some of the concerns. For experimentation, the authors provided 6 additional tables, which is very impressive. But this also suggests that the paper should go through another round of thorough review to validate the massive additional experiments.

---

### Decision · Program_Chairs · 2026-01-26

Reject